# CAMDROP: GRADIENT-GUIDED DYNAMIC FEATURE DROPPING FOR MULTIMODAL BALANCED LEARNING

## ABSTRACT

Multimodal learning seeks to integrate complementary information from diverse modalities to enhance model performance. However, modality imbalance—where dominant modalities overshadow weaker ones—often hinders effective representation learning and generalization. Existing solutions are either gradient-based, which lack fine-grained control, or feature-based, which suffer from randomness and low interpretability. To address these challenges, we propose **CAMDrop**, a lightweight and plug-and-play strategy that suppresses dominant modality regions guided by class activation maps (CAMs). By leveraging GradCAM, CAMDrop identifies class-relevant spatial features and adaptively masks them based on instance-level importance, enabling semantically meaningful and dynamically adjusted suppression without altering model architecture or training objectives. Extensive experiments on three benchmarks, demonstrate that CAMDrop consistently improves accuracy and robustness, effectively mitigating modality imbalance while enhancing model interpretability. Beyond quantitative gains, CAMDrop provides qualitative insights into modality contributions, offering a transparent mechanism for balanced learning. We believe our method can serve as a practical component for future multimodal systems in applications such as emotion recognition, event localization, and human–computer interaction.

## 1 INTRODUCTION

Human perception integrates multiple modalities, such as vision, hearing, and touch, to form a comprehensive understanding of the world Hollier et al. (1999). Inspired by this, multimodal learning aims to fuse information from different sources Ngiam et al. (2011). Traditional machine learning methods often focus on a single modality, ignoring the potential of cross-modal integration Baltrušaitis et al. (2018). With the advancement of sensing technology, it is now feasible to collect and utilize multimodal data for learning tasks Ramachandram & Taylor (2017). In recent years, multimodal learning has demonstrated notable advantages with a variety of tasks, such as action recognition Yu et al. (2023); Kazakos et al. (2019), visual quizzing Ilievski & Feng (2017), and sentiment analysis Gandhi et al. (2023).

Although multimodal models typically outperform unimodal ones, recent studies have shown that their performance often falls short of expectations Wang et al. (2020b). A key reason is modality imbalance, where the model tends to over-rely on the dominant modality during training, leading to under-utilization of the weaker one. This imbalance hinders the model's ability to fully leverage complementary information and is considered a major bottleneck in multimodal learning Zadeh et al. (2017); Zhang et al. (2024).

Existing approaches to alleviate modality imbalance can be broadly categorized into two groups: gradient-based and feature-based methods. Gradient-based methods adjust the backward flow of gradients to reduce reliance on the dominant modality Peng et al. (2022); Wang et al. (2020b); Sun et al. (2021). In contrast, feature-based methods intervene in the forward pass by discarding parts of the feature map corresponding to the stronger modality Wei et al. (2024).

While both strategies show promise, they still have limitations in adaptively balancing the modalities across diverse tasks and instances. (1) Gradient-based methods struggle with coarse and inflexible adjustment. These approaches often reweight gradients globally across layers or modalities, failing to reflect the fine-grained, instance-specific importance of features. As a result, they may

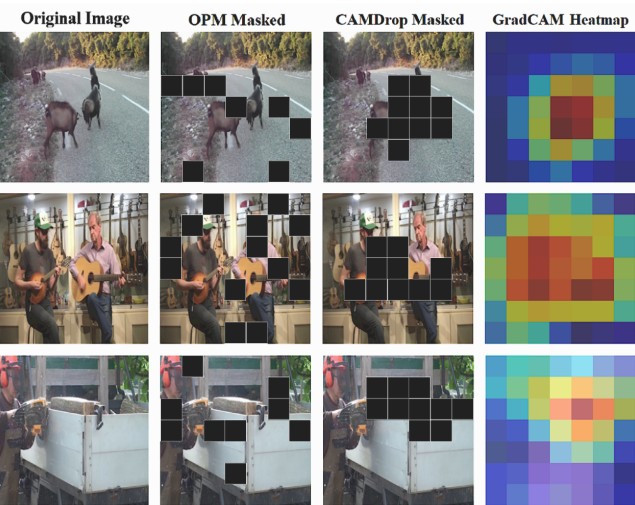

Figure 1: Visual comparison of OPM and CAMDrop on AVE samples with a 0.2 masking ratio. CAMDrop suppresses class-relevant dominant regions more precisely than random OPM, as guided by GradCAM heatmaps.

under-compensate or over-correct when modality dominance varies across samples, limiting their effectiveness in dynamic scenarios He (2024); Wang et al. (2020b). (2) Feature-based methods lack semantic precision and introduce instability. By randomly or uniformly discarding features from the dominant modality, they risk suppressing both informative and uninformative regions alike, making the masking process hard to interpret. Additionally, the stochastic nature of feature dropping increases training variance and hampers convergence Wei et al. (2024); Dai et al. (2024).

To overcome the rigidity of gradient-based strategies and the semantic ambiguity of feature-based masking, we propose Class Activation Map-guided Dropout (**CAMDrop**), a novel approach that introduces instance-aware, class-discriminative feature suppression to address modality imbalance. Class activation maps (CAMs) identify spatial regions that contribute most to a model's prediction by combining gradient information with feature activations. CAMDrop directly tackles the coarse granularity of gradient modulation by localizing where the model over-relies within each modality, and it avoids the randomness and poor interpretability of naive feature dropping by applying semantically meaningful masks. Specifically, CAMDrop leverages GradCAM Selvaraju et al. (2017); Chattopadhay et al. (2018) to generate heatmaps that highlight spatial regions most relevant to the predicted class. This is achieved by computing importance weights for each feature map channel based on the gradients flowing back from the class score, thereby identifying high-activation areas with strong class dependency. These dominant regions are then selectively masked during the forward pass, suppressing overrepresented modality contributions and guiding the model to attend to underutilized signals. Unlike prior approaches Wei et al. (2024) that rely on random or uniform masking, CAMDrop provides a targeted, interpretable, and lightweight solution—requiring no architectural changes or loss function modifications—and can be seamlessly integrated into existing multimodal frameworks Tsai et al. (2019). We validate the effectiveness of CAMDrop through comprehensive experiments on multiple benchmark multimodal datasets. Results demonstrate that CAMDrop not only improves overall performance, but also effectively mitigates modality imbalance, leading to more robust and generalizable multimodal representations.

To summarize, our contributions in this paper are as follows:

- We propose CAMDrop, a novel semantically guided feature masking strategy that combines gradient-based attention and forward-pass feature suppression.

- CAMDrop introduces no architectural modifications or training loss changes, making it lightweight and easily applicable to diverse multimodal frameworks.

- Extensive experiments demonstrate that CAMDrop effectively mitigates modality imbalance and achieves state-of-the-art performance across multiple benchmarks.

## 2 RELATED WORK

### 2.1 MODALITY IMBALANCE IN MULTIMODAL LEARNING

Multimodal learning aims to integrate complementary information across modalities, but performance often suffers from modality imbalance, where models over-rely on the dominant modality Wang et al. (2020b); Zadeh et al. (2017), limiting cross-modal synergy.

To address this, gradient-based methods like OGM Peng et al. (2022) downscale updates from the dominant modality, yet they rely on global statistics and lack fine-grained control. Feature-based approaches such as OPM Peng et al. (2022) and DropPathway Xiao et al. (2020) randomly discard dominant modality features, but are stochastic and semantically uninformed, potentially causing unstable training.

Recent works introduce structured balancing: PMR Zhang et al. (2024) promotes weaker modalities via prototype alignment, MMCosine Xu et al. (2023) enforces balanced representation with contrastive loss, and graph-based GIN Sun et al. (2023) adjusts gradients based on modality interaction. However, these methods often add architectural overhead, need extra supervision, or lack interpretability.

Building on these insights, we propose a semantically guided dropout that suppresses dominant modality signals in a class- and instance-specific manner, avoiding randomness and architectural changes, thus offering interpretable, flexible, and practically integrable multimodal balancing.

### 2.2 CLASS ACTIVATION MAPS

Class Activation Maps (CAMs) highlight spatial regions most relevant to a target class, providing intuitive insight into model decisions Zhou et al. (2016). Later variants such as GradCAM Selvaraju et al. (2017) extend this idea to a broader range of architectures by leveraging the gradient of the class score with respect to feature maps, enabling instance-specific and class-aware localization. Further refinements like GradCAM++ Chattopadhay et al. (2018) and Score-CAM Wang et al. (2020a) improve resolution and stability.

In this work, we choose GradCAM as the foundation for CAMDrop. Compared to the original CAM, GradCAM does not require a dedicated global pooling classifier layer, making it more broadly applicable to diverse multimodal backbones. More importantly, it provides finer, instance-level discriminative cues that are crucial for identifying over-dominant modality regions. While GradCAM introduces an additional backward pass, our empirical study shows the computational overhead is modest, and the gains in accuracy and stability outweigh the cost Rony et al. (2019). These characteristics make GradCAM particularly well suited for our goal of semantically guided, adaptive suppression in multimodal learning.

## 3 METHODS

In this work, we adopt **GradCAM** Selvaraju et al. (2017), a gradient-based class activation mapping technique, as the semantic foundation for identifying modality-dominant regions. GradCAM enables the localization of class-discriminative regions by tracing the gradient of the target class score back to the convolutional feature maps.

Given an input sample and a target class $c$, GradCAM estimates the importance of each feature map channel $A^k$ (of spatial size $H \times W$) in the last convolutional layer with respect to the class score $y^c$. This is done by computing the partial derivative: $\partial y^c / \partial A_{ij}^k$.

The resulting gradients are spatially averaged to obtain a weight coefficient $\alpha_k^c$ for each channel $k$, which reflects how strongly each feature map contributes to the target prediction. The GradCAM heatmap is then computed as a weighted sum of the feature maps with ReLU activation:

$$\alpha_k^c = \frac{1}{H \cdot W} \sum_{i=1}^{H} \sum_{j=1}^{W} \frac{\partial y^c}{\partial A_{ij}^k} \qquad L_{\text{GradCAM}}^c = \text{ReLU}\left( \sum_k \alpha_k^c A^k \right) \qquad (1)$$

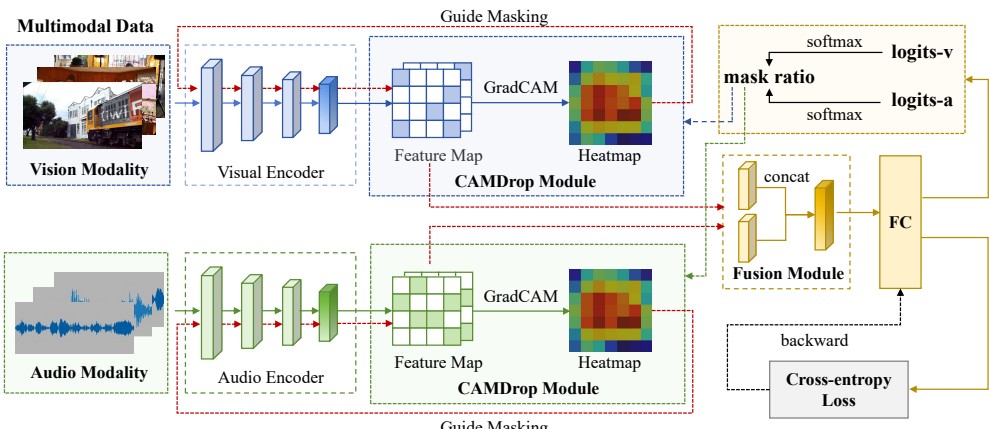

Figure 2: The pipeline of CAMDrop: a GradCAM-guided feature suppression strategy for multi-modal fusion.

The output heatmap $L^c_{\text{GradCAM}}$ highlights spatial regions most influential for predicting class $c$, while preserving spatial alignment for intuitive visualization and downstream semantic guidance.

CAMDrop is a semantically guided feature suppression strategy that leverages class-discriminative activation maps to attenuate the dominance of over-represented modalities in multimodal learning. Unlike prior methods that apply stochastic masking or gradient heuristics, CAMDrop performs instance-aware dropout based on spatial class relevance, thereby improving both interpretability and balancing capacity.

**GradCAM Heatmap Generation for Class-Relevant Localization.** Let $A^{(m)} \in R^{C \times H \times W}$ denote the feature map of modality $m$ from the last convolutional layer used for GradCAM, where $C$ is the number of channels and $H, W$ are the height and width of the spatial dimensions. We first compute the gradient of $y^c$ with respect to each spatial location of the feature map $A^{(m)}$:

$$\frac{\partial y^c}{\partial A^{(m)}_{kij}} \quad \text{for } k = 1, \ldots, C; \ i = 1, \ldots, H; \ j = 1, \ldots, W \tag{2}$$

This gradient measures how sensitive the class score $y^c$ is to changes in the activation at position $(i, j)$ in channel $k$, reflecting the importance of each spatial unit to the prediction. To quantify the overall importance of each channel $k$, we take the average of these gradients over all spatial positions:

$$\alpha^{(m),c}_k = \frac{1}{H \cdot W} \sum_{i=1}^{H} \sum_{j=1}^{W} \left( \frac{\partial y^c}{\partial A^{(m)}_{kij}} \right) \tag{3}$$

This channel weight $\alpha_k$ serves as a global importance indicator of the feature map $k$ for the class $c$.

Using these weights, the final class activation map (CAM) is formed as a weighted combination of the original feature maps:

$$L^{(m),c}_{\text{GradCAM}}(i, j) = \text{ReLU} \left( 0, \sum_{k=1}^{C} \alpha^{(m),c}_k \cdot A^{(m)}_{kij} \right) \tag{4}$$

Here, the ReLU operation ensures that only positively correlated activations contribute, emphasizing regions that support the current class decision. The resulting heatmap $L^{(m),c}_{\text{GradCAM}}$ highlights the spatial areas most responsible for the prediction of class $c$.

**Top-$r$ Region Masking for Dominance Suppression.** Given the heatmap, we apply semantic masking to suppress dominant modality regions. We define a masking ratio $r \in (0, 1)$, representing the proportion of the most activated regions to be suppressed.

To locate the most influential areas, we compute a threshold $T_r^{(m,c)}$ such that exactly $r \cdot H \cdot W$ locations in the CAM have activations above this threshold:

$$\left| \left\{ (i,j) \mid L_{\text{GradCAM}}^{(m),c}(i,j) > T_r^{(m,c)} \right\} \right| = r \cdot H \cdot W \tag{5}$$

This defines a per-sample, per-modality cutoff for what is considered "too dominant". We then construct a binary mask $M^{(m,c)}$ that retains only the non-dominant areas:

$$M^{(m,c)}(i,j) = \left\{ \begin{array}{ll} 0, & \text{if } L_{\text{GradCAM}}^{(m),c}(i,j) > T_r^{(m,c)} \\ 1, & \text{otherwise} \end{array} \right. \tag{6}$$

In this mask, locations with strong class-related responses (i.e., above the threshold) are zeroed out (dropped), while less dominant areas are preserved. Alternatively, this mask can be written in indicator function form:

$$M^{(m,c)}(i,j) = \left[ L_{\text{GradCAM}}^{(m),c}(i,j) \leq T_r^{(m,c)} \right] \tag{7}$$

To determine the masking ratio $r$ dynamically for each modality, we assess modality dominance. For a batch of size $B$, let $z_i^{(v)}$, $z_i^{(a)}$ be the logits from the visual and audio encoders for the $i$-th sample, and $c_i$ its true class. The softmax confidence for the correct class is:

$$y_i^{(m)}[c] = \frac{\exp(z_i^{(m)}[c])}{\sum_{k=1}^{K} \exp(z_i^{(m)}[k])} \tag{8}$$

We then define the dominance ratio of the visual modality as:

$$\rho^{(v)} = \frac{1}{B} \sum_{i=1}^{B} \frac{y_i^{(v)}[c_i]}{y_i^{(a)}[c_i]}, \quad \rho^{(a)} = \frac{1}{\rho^{(v)}} \tag{9}$$

Here, $B$ is the batch size, and $c_i$ is the ground-truth class index for the $i$-th sample. The scores $y_i^{(v)}$ and $y_i^{(a)}$ represent the softmax output vectors from the visual and audio branches, respectively.

The final masking ratio $r^{(m)}$ for modality $m$ is computed as:

$$r^{(m)} = \sigma \left( \log \left( \rho^{(m)} \right) \right) \cdot r_{\max} \tag{10}$$

where $\sigma(x) = \frac{1}{1+\exp(-x)}$ is the sigmoid function, and $r_{\max}$ is the user-defined maximum masking ratio (e.g., 0.3). This adaptive strategy enables CAMDrop to scale the suppression strength according to the relative dominance between modalities.

**Mask Application to Feature Maps.** The binary mask $M^{(m,c)} \in \{0,1\}^{H \times W}$ is broadcast along the channel axis of the feature map $A^{(m)}$ to form the masked output:

$$\tilde{A}_{kij}^{(m)} = A_{kij}^{(m)} \cdot M_{ij}^{(m,c)} \tag{11}$$

This operation sets to zero those spatial locations identified as overly dominant, forcing the model to rely on underused but relevant regions. Unlike traditional dropout, this suppression is semantic and deterministic, improving interpretability and training consistency.

**Logit Computation and Fusion.** For each modality, we obtain pooled feature representations $f^{(v)}$ and $f^{(a)}$ from the feature maps $A^{(v)}$ and $A^{(a)}$, which are passed through lightweight linear classifiers to produce unimodal logits $z^{(v)}$ and $z^{(a)}$. These auxiliary logits are used solely for computing the dominance ratio (Eq. 7) and do not affect the main prediction. After masking, the modality features $\tilde{A}^{(v)}$ and $\tilde{A}^{(a)}$ are fused (e.g., by concatenation) and fed into a classifier to obtain the fused logits $z$, which are used for the final cross-entropy loss. Notably, CAMDrop introduces no new parameters or loss terms and integrates seamlessly into existing pipelines while reducing training variance and improving modality balance.

## 4 EXPERIMENTS

### 4.1 DATASETS

**CREMA-D** Cao et al. (2014) is an audio-visual dataset designed for emotion recognition research, featuring both facial expressions and vocal cues. It categorizes emotions into six types: happiness, sadness, anger, fear, disgust, and neutrality. The dataset contains a total of 7,442 audio-visual clips, which are randomly split into a training set of 6,698 samples and a test set of 744 samples.
**AVE** Tian et al. (2018) is an audio-visual dataset for audio-visual event localization. It comprises 4,143 video clips, each 10 seconds long, containing synchronized visual and auditory streams with frame-level event annotations. The dataset spans 28 event categories, and all videos are sourced from YouTube. For experiments, the dataset is partitioned following the described protocol.

**Kinetics-Sounds** (KS) Arandjelovic & Zisserman (2017) is a subset of the Kinetics-400 dataset Kay et al. (2017), consisting of 31 human action classes selected for their clear audio-visual correspondence. Each video is sourced from YouTube and is manually annotated for human actions via Amazon Mechanical Turk. Videos are trimmed to 10-second clips centered around the labeled actions. The selected classes are those likely to exhibit both visual and auditory cues, such as playing instruments, laughing, or singing. In total, the dataset comprises approximately 19,000 10-second clips, with a split of 15,000 for training, 1,900 for validation, and 1,900 for testing.

### 4.2 EXPERIMENTAL SETTINGS

We employ a ResNet18-based architecture He et al. (2016) for both visual and audio modalities. The visual encoder processes 2D frames sampled at 1 fps; for AVE and Kinetics-Sounds, 3 frames are uniformly selected from each 10-second video. For CREMA-D, a single frame is used due to its shorter clip length. The audio encoder modifies the first ResNet18 layer to accept single-channel input, following Chen et al. (2020), while preserving the rest of the structure. All audio clips are converted into log-mel spectrograms using librosa McFee et al. (2015), with a window length of 512 and hop length of 159, resulting in spectrograms of size $257 \times 1004$ (AVE/KS) and $257 \times 299$ (CREMA-D). We optimize the model using SGD with a momentum of 0.9, weight decay of $1 \times 10^{-4}$, and an initial learning rate of $1 \times 10^{-3}$ that decays by 0.1 every 70 epochs. For CAMDrop, the maximum masking ratio $r_{max}$ is set to 0.3. To avoid noisy GradCAM masks at the beginning of training, CAMDrop is activated with a small initial masking ratio of 0.25 and then linearly increased to the target $r_{\max}$ within the first 10 epochs. This warm-up stabilizes optimization while still allowing effective suppression.

### 4.3 COMPARISON ON THE MULTIMODAL TASK

**Comparison across fusion strategies and modulation methods.** To provide a comprehensive evaluation, we compare CAMDrop with state-of-the-art modulation strategies (Gradient-Blending Wang et al. (2020b), OGM-GE Peng et al. (2022), OPM Wei et al. (2024), PMR Fan et al. (2023)) under three widely adopted fusion schemes: concatenation, summation, and FiLM Perez et al. (2018). Table 1 summarizes the results on CREMA-D, AVE and KS datasets.

We observe that CAMDrop consistently improves performance across all fusion strategies, with the best results achieved when $r_{\max} = 0.3$. This demonstrates the effectiveness of semantically guided masking in balancing modality contributions while preserving discriminative information. Notably, CAMDrop outperforms OGM-GE and OPM in most settings, despite not requiring additional parameters, classifiers, or heuristic selection strategies.

| Model | Method | CREMA-D | | | AVE | | | KS | | |
|---|---|---|---|---|---|---|---|---|---|---|
| | | Concat | Sum | FiLM | Concat | Sum | FiLM | Concat | Sum | FiLM |
| Baseline | Naive Fusion | 57.8 | 57.5 | 59.9 | 60.7 | 60.2 | 62.4 | 50.4 | 50.5 | 49.4 |
| | Grad-Blending | 59.7 | 58.6 | 59.1 | 63.9 | 62.2 | 62.7 | 53.0 | 52.2 | 49.1 |
| | OGM-GE | 59.7 | 62.0 | 59.6 | 62.4 | 61.4 | 62.1 | 51.3 | 52.6 | 52.3 |
| | OPM | 60.3 | 61.2 | 61.2 | 61.9 | 60.7 | 63.2 | 51.0 | 51.7 | 52.5 |
| | PMR$^{\ddagger}$ | 61.1 | 59.4 | 61.8 | - | - | - | - | - | - |
| CAMDrop | $r_{max}$=0.1 | 61.7 | 62.5 | 62.9 | **65.4** | 64.0 | 63.4 | 52.7 | 52.4 | 52.3 |
| | $r_{max}$=0.3 | **63.8** | **63.6** | **63.3** | 64.9 | 63.9 | **64.2** | 52.9 | **53.6** | 53.0 |
| | $r_{max}$=0.5 | 62.0 | 61.6 | 62.2 | 64.4 | 63.4 | 63.2 | **53.1** | 52.4 | **53.4** |
| Uni-modal | Audio-only | | 57.3 | | | 55.0 | | | 30.1 | |
| | Visual-only | | 29.0 | | | 33.6 | | | 29.9 | |

Table 1: Accuracy comparison across different fusion strategies and modulation methods on CREMA-D, AVE, and KS datasets. CAMDrop consistently outperforms other baselines under all settings, with $r_{max}$=0.3 showing the best trade-off between suppression strength and feature preservation. $^{\ddagger}$Results for PMR are directly reported from the original paper Fan et al. (2023).

| Method | CREMA-D | AVE | KS |
|---|---|---|---|
| TBN | 57.1 | 57.5 | 46.5 |
| PSP | 61.3 | 60.0 | 49.9 |
| TBN$^{\dagger}$ | 59.9 | 59.0 | 48.4 |
| PSP$^{\dagger}$ | **61.5** | **63.2** | **53.8** |

Table 2: Performance comparison of CAMDrop integrated with multimodal architectures. $^{\dagger}$ indicates CAMDrop is applied.

While parameterized fusion methods like FiLM already adapt modality contributions, CAMDrop still provides further gains, indicating that our approach brings complementary benefits by explicitly regulating feature dominance. These results validate CAMDrop's flexibility and robustness, making it suitable for diverse fusion architectures and learning scenarios.

**Combination with existing methods.** To assess the flexibility and effectiveness of our proposed CAMDrop, we integrate it into two representative baseline methods: TBN Kazakos et al. (2019), a classical two-stream fusion model, and PSP Zhou et al. (2021), a progressive multi-stage interaction network for audiovisual understanding.

As shown in Table 2, these results collectively demonstrate that CAMDrop is not limited to naive fusion scenarios, but can also benefit from more sophisticated fusion architectures. Its ability to generalize across different multimodal designs—whether simple or deeply coupled—highlights CAMDrop's practicality and adaptability for a wide range of multimodal learning frameworks.

## 4.4 PARAMETER SENSITIVITY

**Impact of masking ratio.** To investigate the sensitivity of CAMDrop to its key hyperparameter, the maximum masking ratio $r_{max}$, we conduct experiments on CREMA-D and AVE datasets by varying $r_{max}$ in the range $\{0.1, 0.2, 0.3, 0.4, 0.5\}$. The results are illustrated in Fig. 3. For CREMA-D, the accuracy improves as $r_{max}$ increases from 0.1 to 0.3, reaching a peak of 63.8%, and then slightly decreases as more class-relevant features are removed. A similar trend is observed on AVE, where the performance peaks at a lower $r_{max} = 0.1$ with 65.4% accuracy and gradually drops afterward. These findings indicate that while CAMDrop is generally robust across a range of masking ratios, optimal performance is achieved when $r_{max}$ strikes a balance between dominant modality suppression and information preservation.

**Adaptive optimizers.** To evaluate the adaptability of our proposed CAMDrop method to different optimization strategies, we integrate CAMDrop into models trained with the widely used Adam optimizer in addition to the default SGD setup. As shown in Table 3, applying CAMDrop leads to

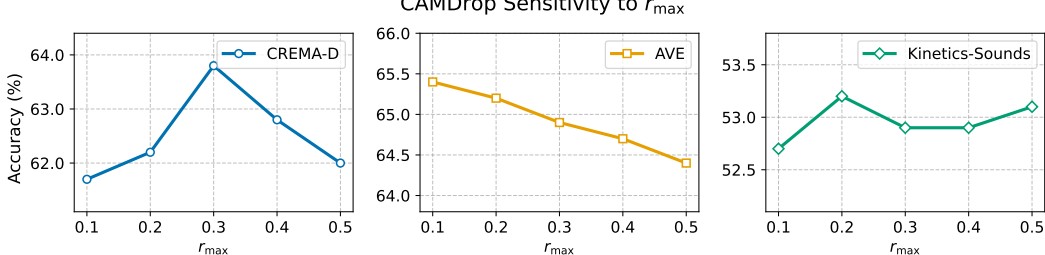

Figure 3: Accuracy variation of CAMDrop on different datasets with respect to masking ratio $r_{max}$.

significant performance improvements for both optimizers compared to their respective baselines. Furthermore, under the same experimental conditions, models trained with SGD achieve slightly better performance than those using Adam. These results demonstrate that CAMDrop is compatible with various optimizers and can consistently enhance model performance regardless of the chosen optimization scheme. These consistent improvements across different optimizers further verify the generality of CAMDrop beyond specific training settings.

| Optimizer | CREMA-D | AVE | KS |
|---|---|---|---|
| SGD | 57.8 | 60.7 | 50.4 |
| Adam | 58.6 | 61.2 | 50.6 |
| SGD† | **63.8** | **64.9** | 52.9 |
| Adam† | 62.8 | **64.9** | **53.7** |

Table 3: Performance comparison using different optimizers with and without CAMDrop.

| Settings | CREMA-D | AVE | KS |
|---|---|---|---|
| (b=16, lr=1e-3) | 63.8 | 64.9 | 52.9 |
| (b=64, lr=1e-3) | 59.8 | 61.9 | 51.7 |
| (b=16, lr=1e-3) | 63.8 | 64.9 | 52.9 |
| (b=16, lr=5e-4) | 61.0 | 64.4 | 52.6 |
| (b=16, lr=1e-4) | 55.8 | 56.0 | 52.3 |

Table 4: Effect of batch size and learning rate configurations on CAMDrop.

**Impact of batch size and learning rate.** To further understand the robustness and generalization behavior of CAMDrop, we conduct a study analyzing the effect of different optimization configurations. Following insights from stochastic optimization theory, the intensity of gradient noise introduced by SGD is influenced by the ratio of learning rate to batch size. Larger learning rates or smaller batches are generally associated with higher noise levels, which may help in escaping sharp minima and improving generalization.

As shown in Table 4, using a smaller batch size with a relatively large learning rate ($b = 16, lr = 1e-3$) achieves the best performance, supporting the hypothesis that stronger gradient noise enhances generalization. As batch size increases or learning rate decreases, model performance drops notably, especially under low learning rates. This indicates that CAMDrop benefits from optimization settings introducing sufficient stochasticity, emphasizing the importance of tuning hyperparameters to fully leverage its potential.

### 4.5 ABLATION STUDY

**Uni-modal performance comparison.** As we discussed before, dominance from one modality in multimodal fusion can hinder the representation learning of the weaker modality. To explore this phenomenon, we compare the uni-modal performance of two variants: (1) uni-modal branches within standard multimodal fusion models, and (2) the uni-modal branches after applying the CAMDrop strategy.

As shown in Fig. 4a,b, the visual branch exhibits performance improvement after applying CAMDrop, whereas the audio branch undergoes degradation compared to the baseline. This observation highlights CAMDrop's design principle: by semantically suppressing class-irrelevant yet over-dominant cues from the strong modality (audio), the weaker modality (visual) better learns discriminative representations. Although this may lead to marginal sacrifice in the dominant modality's performance, the effect is a more balanced and complementary fusion. Notably, the enhanced visual learning contributes to improved final classification, suggesting that recovering the weak modality is key to unlocking multimodal synergy. This trade-off is especially beneficial in real-world scenarios

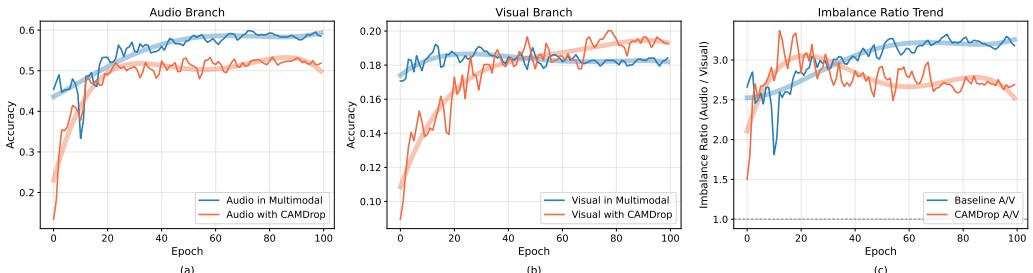

Figure 4: Uni-modal performance and imbalance ratio analysis. (a) and (b) show the classification accuracy of the audio and visual branches within the multimodal model, respectively, comparing standard fusion baseline and CAMDrop-enhanced models. While the audio branch experiences a slight drop after applying CAMDrop, the visual branch shows a substantial improvement, indicating better feature exploitation of the weaker modality. (c) tracks the imbalance ratio $\rho$ (i.e., audio accuracy / visual accuracy) over training epochs. CAMDrop consistently reduces $\rho$, demonstrating its ability to alleviate modality imbalance and encourage more balanced representation learning.

where modalities vary in quality or informativeness. Furthermore, CAMDrop avoids manual tuning of fusion weights, offering a more adaptive, data-driven balancing strategy.

To further quantify this effect, we track the modality imbalance during training using an *imbalance ratio* $\rho$, which measures the relative performance between audio and visual branches. As shown in Fig. 4c, CAMDrop consistently reduces $\rho$ throughout training, indicating a more equitable learning process. The gap between modalities shrinks after the dominant modality has plateaued, suggesting that CAMDrop continues to encourage the weaker modality's learning without relying on heuristic re-weighting or auxiliary supervision. These results affirm that CAMDrop not only enhances visual representation but also promotes multimodal synergy through targeted suppression.

## 5 CONCLUSION & DISCUSSION

CAMDrop is a simple yet effective masking strategy that semantically suppresses dominant modality cues to enhance multimodal balance. It introduces no additional parameters or branches and generalizes well across multiple datasets and fusion strategies, consistently improving performance. These results highlight the potential of class-guided suppression in multimodal learning and underscore its practicality in real-world settings with modality imbalance. Furthermore, CAMDrop's architecture-agnostic design makes it easily integrable into a wide range of multimodal pipelines.

While CAMDrop demonstrates clear advantages in addressing modality imbalance, several aspects merit further discussion.

**Interpretability and Stability.** CAMDrop's deterministic and interpretable masking identifies semantically meaningful regions for suppression, contributing to stable training. This property facilitates debugging and model understanding, especially in safety-critical applications where transparency is critical.

**Hyperparameter Sensitivity.** Our results show that the masking ratio $r_{\max}$ influences performance. Moderate values (e.g., 0.3) offer optimal balance, while excessive masking may harm useful feature retention. Future extensions could explore adaptive or learnable masking mechanisms to improve robustness and reduce manual tuning effort.

**Limitations.** CAMDrop relies on GradCAM to estimate class-relevant regions, which introduces an extra backward pass. Additionally, the quality of these heatmaps may vary with network architecture or task supervision, potentially affecting suppression accuracy.

**Future Directions.** CAMDrop can be further extended with advanced CAM variants (e.g., Grad-CAM++, Score-CAM), or adapted to temporal masking for video/audio tasks. Applying it to transformer-based multimodal models is another promising direction, especially given their growing prevalence and fine-grained attention mechanisms.

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

## A    APPENDIX

You may include other additional sections here.

