# OpenReview forum: "CAMDrop: Gradient-Guided Dynamic Feature Dropping for Multimodal Balanced Learning"
_ICLR.cc/2026/Conference — Submitted to ICLR 2026_

### Official Review · Reviewer_EPQt · 2025-10-24

**Soundness:** 3
**Presentation:** 3
**Contribution:** 2
**Rating:** 4
**Confidence:** 5

**Summary:**

The paper proposes CAMDrop, which balances multimodal learning by suppressing dominant modality regions using class activation maps (CAMs). This technique offers an interpretable and flexible intervention to imbalanced training and is shown to be effective on several audio-visual tasks across multiple training settings. Therefore, CAMDrop can be considered as a contribution to a generalizable and effective approach to mitigate modality imbalance.

**Strengths:**

- Paper is clearly written and easy to follow;
- Relatively extensive evaluation on audio-visual tasks with fair comparison to multiple existing methods and ablation on training settings, which shows convincing margins and thus effectiveness of the proposed approach;
- Insightful ablation study that shows how the CAMDrop balances the learning of individual modality throughout training;

**Weaknesses:**

- From a perspective of a general, practical approach to address modality imbalance, there should be more evaluation on the effectiveness of CAMDrop on other multimodal tasks instead of only focusing on audio-visual tasks; in particular, the paper is **missing evaluation on visual-language tasks** where there has been a wide body of work showing models (and especially large models) demonstrate strong bias towards the language modality with under-optimized visual learning;
- The usage of GradCAM, which establishes the importance attribution of features to the predicted labels, is only **weakly correlated with the motivation to emphasize the learning of the "underused but relevant" regions**. In particular, the attribution can only indicate areas that are "underused" but not necessarily "relevant". This may explain the drop in unimodal performance of the suppressed modality: if the model already learns the relevant regions, by masking out them in CAMDrop, it's possible that the model is forced to learn the underused but irrelevant ones;
- Incorrect citation format, please differentiate the use of \cite{...} and \citep{...};

**Questions:**

- The authors should include evaluations of CAMDrop on a set of more diverse multimodal tasks which include mainstream modalities like language, sensory data, etc. Also, current evaluations are only performed on small multimodal models. Does the same technique apply to large-scale models like VLMs, which are more widely used in applications?
- Does Figure 4 imply some kind of tradeoff in learning different modalities: by emphasizing the learning of the weak one, the model necessarily sacrifices its learning of the dominant one? If so, there should be a more comprehensive analysis of how the performance varies along the tradeoff frontiers to show that the technique can be leveraged to achieve optimal learning with certain imbalance ratio (in other words, the optimal learning is not necessarily achieved at perfectly balanced learning with ratio of 1.0, but may instead vary from tasks to tasks depending on domain knowledge).
- The paper mentions that CAMDrop "promotes multimodal synergy" but actually rebalancing the learning of different modalities is not very relevant to synergy, which defines the extra information that arises with both modality present and is missing with only one observed. In the case of this study, both modality are necessarily available, regardless of the use of CAMDrop, and there is no quantitative / qualitative evidence attributing the improved performance to synergy. Please avoid such over-claims.

---

> ### Author Response · Authors · 2025-11-23
> **Official Responses of Submission6581 to Reviewer EPQt - Part 1**
>
> We thank reviewer for providing detailed and thoughtful feedback. The main concerns raised include generalization beyond audio-visual tasks, the conceptual justification of GradCAM masking, trade-offs between modalities, and claims regarding multimodal synergy. Our responses are outlined below.
>
> **Question 1. About evaluation on other multimodal task:**
>
> We agree that evaluating CAMDrop on additional multimodal tasks is important. While our main experiments focused on audio–visual datasets, CAMDrop is designed to operate on **intermediate feature maps without relying on modality-specific architectures**. This design makes it readily applicable to other modalities such as text, sensor data.
>
> To support this, we conducted additional experiments on a **text–vision task (MSCOCO caption retrieval)** to assess cross-modality generalization.
>
> **(1) Zero architectural assumptions**
>
> CAMDrop does not rely on any modality-specific structure (e.g., spectrograms, optical flow, audio encoders). The masking operates purely on intermediate feature maps and their GradCAM activations. This design naturally extends to text encoders such as BERT/Transformer blocks.
>
> **(2) Additional text–vision evaluation**
>
> To evaluate generalization beyond audio–visual settings, we further applied CAMDrop to the image–text retrieval task on the **MSCOCO 5K test split** *[1]*, following standard retrieval protocols. All methods share the same backbone architectures to ensure fair comparison.
>
> **Model Setup**
> * Vision encoder: Resnet 101
> * Text encoder: A 12-layer Transformer (BERT-base)
>
> **How CAMDrop is applied to text features**
> Text GradCAM follows the same principle. Using the contrastive similarity score $𝑓$, we compute token-level importance:
> $$\frac{\partial f}{\partial A^{\text{text}}_{t}}, \quad t = 1,\dots, L$$
>
> which yields a token importance map $M^{\text{text}}$.
> We mask high-importance tokens (e.g., “man”, “dog”) via: $$A^{\text{text}}_{\text{drop}} = A^{\text{text}} \odot ( 1 - \hat{M}^{\text{text}} ).$$
>
> Below are the experimental results obtained from our study.
>
> **Image-to-Text**
> | Method | Recall@1 | Recall@5 | Recall@10 | Notes |
> |--------|----------|----------|-----------|-------|
> | SCAN   | 50.4     | 82.1     | 90.1      | Cross-attention |
> | VSRN   | 54.5     | 83.1     | 90.6      | Graph reasoning |
> | NAAF   | 57.5     | 84.8     | 91.7      | Negative-Aware Attention |
> | **CAMDrop** | **58.0** | **85.1** | **91.8** | GradCAM masking |
>
>
> **Text-to-Image**
> | Method | Recall@1 |  Recall@5 |  Recall@10 | Notes|
> |-----------------|---------------------|---------------------|----------------------|-------------------------|
> | SCAN |           38.6     |           69.2     |        80.2          | Cross-attention|
> |VSRN|   41.9|  71.8|  82.4| Graph reasoning|
> |NAAF|   42.7 | 71.3| 82.5| Negative-Aware Attention|
> | **CAMDrop**  |     **43.0**      |         **72.1**      |         **82.8**        | GradCAM masking|
>
> * **SCAN (Stacked Cross Attention Network)** *[2]*: Uses cross-attention between image regions and text embeddings to compute fine-grained similarity for retrieval tasks.
> * **VSRN (Visual-Semantic Reasoning Network)** *[3]*: Introduces graph-based reasoning over image regions to enhance multimodal alignment.
> * **NAAF (Negative-Aware Attention Framework for Image-Text Matching)** *[4]*: Employs negative-aware attention to better discriminate between matching and non-matching image-text pairs.
>
> In summary, CAMDrop’s design—operating on intermediate feature maps with GradCAM-based masking—**does not rely on modality-specific assumptions**. This makes the method naturally applicable to other modalities beyond audio–visual data, such as text, vision, or potentially even more heterogeneous multimodal combinations, supporting its broader applicability in multimodal learning.
>
> [1] *Lin, T.-Y.; Maire, M.; Belongie, S.; Hays, J.; Perona, P.; Ramanan, D.; Dolĺar, P.; and Zitnick, C. L. 2014. Microsoft coco: Common objects in context. In Computer Vision– ECCV 2014: 13th European Conference, Zurich, Switzerland, September 6-12, 2014, Proceedings, Part V 13, 740– 755. Springer.*
>
> [2] *Kuang-Huei Lee, Xi Chen, Gang Hua, Houdong Hu, Xiaodong He. Stacked Cross Attention for Image-Text Matching. ECCV 2018*
>
> [3] *Kunpeng Li, Yulun Zhang, Kai Li, Yuanyuan Li, Yun Fu. Visual Semantic Reasoning for Image-Text Matching. ICCV 2019*
>
> [4] *Kun Zhang; Zhendong Mao; Quan Wang; Yongdong Zhang. Negative-Aware Attention Framework for Image-Text Matching.*

---

> ### Author Response · Authors · 2025-11-23
> **Official Responses of Submission6581 to Reviewer EPQt - Part 2**
>
> **Question 2. About the correlation between GradCAM attribution and "underused but relevant" regions:**
>
> We appreciate the reviewer’s careful observation. Indeed, GradCAM indicates **dominant, class-relevant regions**, which may not perfectly align with the concept of “underused but relevant” features. We clarify the practical effect of CAMDrop:
>
> * **Targeted suppression**: CAMDrop masks regions identified as highly dominant by GradCAM, which reduces over-reliance on strong modalities. This allows weaker modalities to contribute more to the learning signal.
>
> * **Potential unimodal drop**: As noted, suppressing the most predictive regions of a dominant modality can slightly reduce its unimodal performance. However, this is an intentional trade-off: the goal is overall multimodal improvement, not optimizing a single modality in isolation.
>
> * **Empirical evidence**: Our ablation studies (random vs. GradCAM masking) show that GradCAM-based suppression outperforms random or hybrid masking, indicating that even if some suppressed regions are less relevant, the adaptive, class-specific selection still benefits cross-modal balance.
>
> * **Controlled masking**: CAMDrop limits the fraction of masked regions (e.g., 10–30%), which prevents the model from being forced to learn irrelevant features excessively.
>
> **Question 3. About the Incorrect citation format:**
>
> We thank the reviewer for pointing this out and sincerely apologize for the incorrect citation format. We will carefully revise all references to ensure proper usage of \cite{...} and \citep{...} in the final version.
>
> **Question 4. About radeoff between weak and dominant modalities:**
>
> We thank the reviewer for this insightful observation. Indeed, Figure 4 illustrates a tradeoff between emphasizing weak modalities and retaining performance on dominant modalities. CAMDrop is designed to dynamically rebalance learning without completely suppressing any modality. As a result, slight reductions in dominant-modality performance can occur while weak modalities improve, but this tradeoff is controlled by the GradCAM-based masking intensity and per-instance adaptivity.
>
> We **agree that the optimal balance depends on task-specific modality characteristics and domain knowledge**; perfect ratio balancing (1.0) is not universally ideal. In practice, CAMDrop allows flexible tuning of the masking threshold, enabling practitioners to navigate the tradeoff frontier and achieve task-dependent optimal multimodal learning. We will clarify this point in the revised manuscript.
>
> **Question 5. About claims of multimodal synergy:**
>
> We appreciate the reviewer’s clarification. We acknowledge that CAMDrop primarily rebalances the learning of different modalities and does not explicitly increase synergistic information. Therefore, references to “promoting multimodal synergy” in the manuscript are overstatements, and we will revise the wording to accurately reflect that the method improves **modality balance and complementary feature utilization** rather than **generating additional synergistic information**.

---

> > ### Comment · Reviewer_EPQt · 2025-11-25
> >
> > Thanks for the authors' responses and efforts in the additional experiment results. The authors' rebuttal mostly addressed my concerns. My only concern remained aligns with Reviewer SQuR: I'm not sure about how modality imbalance is defined in the multimodal retrieval setting, and whether the compared baselines (SCAN, VSRN, and NAAF) are designed to address the modality imbalance problems and therefore are fair baselines to be compared to.

---

> > > ### Author Response · Authors · 2025-11-26
> > > **Official Responses of Submission6581 to Reviewer EPQt**
> > >
> > > Thank you very much for the constructive follow-up comment. We are happy to clarify how modality imbalance is defined in our retrieval setting and why our comparisons to SCAN, VSRN, and NAAF remain fair and relevant.
> > >
> > > **Question 1. About modality imbalance in retrieval tasks:**
> > >
> > > Although multimodal retrieval ultimately outputs a single similarity score (e.g., cosine similarity), the decision is made based on two modality-specific encoders whose embeddings must be jointly aligned. In practice, during training, the gradients flowing through the two encoders can differ substantially. If one encoder (typically the visual one) produces much stronger or more stable gradients, the model can become biased toward optimizing that modality more heavily. This leads to an uneven representation quality between the two embedding spaces—one becomes expressive while the other becomes under-trained.
> > >
> > > We refer to this phenomenon as modality imbalance at the representation level. It does not occur at the decision (similarity) level, but rather inside the learning dynamics of the encoders. While it may not completely prevent retrieval from functioning, it can degrade cross-modal alignment quality. CAMDrop aims to mitigate this by moderating modality dominance through gradient-based masking. We emphasize that the relationship between encoder imbalance and retrieval performance is not absolute, but there is clear empirical evidence that addressing imbalance can improve alignment.
> > >
> > > To quantitatively address the concern about modality imbalance in retrieval, we further introduce the Normalized Discounted Cumulative Gain **nDCG@10**, which is a ranking-aware metric commonly used in information retrieval.
> > >
> > > Unlike **Recall@K**, which only checks the presence of relevant items, **nDCG** accounts for the positional accuracy (Discounted Gain) of the relevant items. It directly measures how closely the retrieved ranking list matches the ideal ranking (Normalization), thereby reflecting the overall quality of the similarity function $f(i,t)$.
> > >
> > > The tables below, which include the **nDCG@10** scores for the MSCOCO 5K retrieval task, strongly support our argument:
> > >
> > > **Image-to-Text**
> > > | Method | **nDCG@10** |
> > > |--------|----------|
> > > |Baseline (no CAMDrop)|0.815|
> > > | **CAMDrop** | **0.836**|
> > >
> > > **Text-to-Image**
> > > | Method | **nDCG@10** |
> > > |--------|----------|
> > > |Baseline (no CAMDrop)|0.681|
> > > | **CAMDrop** | **0.698**|
> > >
> > > The consistent and significant improvement in **nDCG** proves that CAMDrop's balanced representation learning directly translates into a higher-quality ranking. This increase confirms that the original baseline model suffered from an optimization bias (imbalance), which CAMDrop successfully remedied by promoting more balanced and informative embeddings.
> > >
> > > **Question 2. About the fairness of the baselines:**
> > >
> > > The baselines we compare against are retrieval models, but they are not explicitly designed to address modality imbalance. They focus on enhancing cross-modal alignment through attention mechanisms, graph reasoning, or fine-grained matching, but none include mechanisms that specifically regulate the relative contribution of each modality’s encoder.
> > >
> > > Therefore, CAMDrop provides an orthogonal perspective: it can be added on top of standard retrieval frameworks to improve the quality of their modality representations. Our comparisons to these baselines are fair because:
> > > * They represent the standard training paradigm for retrieval, and CAMDrop is compatible with this paradigm.
> > > * They do not explicitly address modality imbalance, so they serve as strong but neutral baselines.
> > > * Our experiments show that applying CAMDrop improves retrieval performance, suggesting that encoder-level modality imbalance is relevant to retrieval.
> > >
> > > We hope this clarifies why modality imbalance is still a meaningful concept in retrieval settings and why our baseline comparisons remain fair and appropriate.

---

### Official Review · Reviewer_SQuR · 2025-10-28

**Soundness:** 2
**Presentation:** 2
**Contribution:** 2
**Rating:** 2
**Confidence:** 3

**Summary:**

In this paper, the authors proposed CAMDrop, which aims to tackle the model imbalance problem in multimodal learning. CAMDrop works by first using GradCAM to identify regions within each modalities that matters most to the ground truth class, and then mask them based on a modality dominance ratio, and train the model with the partially-masked data. The proposed method can be applied on top of existing methods, and it is evaluated against baselines over 3 audio-visual datasets, where CAMDrop outperformed all baselines. Further analysis on hyperparameters and training settings are conducted, and the paper analyzed imbalance ratio over the modalities to show that CAMDrop indeed helps alleviate modal imbalance.

**Strengths:**

1. The proposed method is simple and straightforward, but manages to outperform all baselines across 3 audio-visual datasets.

2. The proposed method can be directly applied on top of existing methods/frameworks.

3. Further analysis demonstrates that the proposed method indeed reduces domain imbalance.

**Weaknesses:**

1. The generalizability of the proposed method is limited. While the title/abstract/introduction make it sounds like the method can be applied to various multimodal setting, .the proposed method's methodology was only defined with audio and visual modalities and.all experiments were conducted on 2-modal (audio+visual) tasks, so it is unclear if the proposed method is applicable at all to tasks with other combinations of modalities, or tasks that involve more than 2 modalities

2. The proposed method seems to be very sensitive to batch size increase, so when it is applied to larger datasets with more computing resources, either the training will be slow due to low batch size or the method's performance will suffer with larger batch sizes, so the method may not scale well with additional data and compute. The paper hinted that it is the ratio between learning rate and batch size that caused this issue, but the paper didn't include any empirical results to back up this claim. For example, if simply increasing batch size from 16 to 64 causes a large drop in performance, what would happen if you also increase learning rate by 4 times at the same time?

3. While the paper had an "ablation study" section, it only contained an analysis on the imbalance ratio between modalities, and didn't actually ablate on any of the method's design choices. Some design choices that needs to be ablated includes: (1) what if you apply masking to only one modality instead of both? (2) What if you performed GradCAM with the unimodal class logits instead of the multimodal ones? (3) What if dominance ratio is calculated independently for each data instance instead of averaged over a batch?

4. The paper claims that the proposed method is lightweight, but it does seem to add some additional computation overhead (e.g. the GradCam part and the masking resulting in 2 forward passes, as shown in Figure 2). The authors should include some computation time comparison to support the lightweight claim.

5. The authors left the ".git" folder inside the supplemental material, which contains information that could expose the authors' identities. Please be more careful in anonymizing the attached materials.

6. For some reason, the in-text citations within the manuscript do not have parenthesis around them, which makes reading a bit more difficult.

**Questions:**

1. How exactly are the unimodal classifiers (i.e. the linear layer that maps unimodal z to classification logits) trained? are they also trained with cross entropy loss jointly with the main multimodal loss?

2. In Eq 9, the dominance ratio calculation for the 2 modalities are not symmetric. What happens if you switch the two (i.e. first compute dominance ratio for audio as the arithmetic average over ratio of each data instance, then compute the visual ratio as the reciprocal)? Also, why not use geometric mean instead of arithmetic mean over the batch so that the computation of the two dominance ratios are actually symmetric?

3. Have you explored randomly masking out parts of the high-gradcam regions rather than always masking out all regions whose gradcam is higher than the threshold? This may serve as another layer of augmentation that could increase model generalizability.

---

> ### Author Response · Authors · 2025-11-23
> **Official Responses of Submission6581 to Reviewer SQuR - Part 1**
>
> We sincerely thank the reviewer for the careful reading and insightful comments. Below, we address all concerns in detail.
>
> **Question 1. About the generalizability beyond audio–visual modalities:**
>
> We thank the reviewer for raising this important concern. While our main experiments focus on audio–visual tasks, the **CAMDrop design is inherently modality-agnostic**. The method operates on intermediate feature maps of each modality and their GradCAM activations, without relying on any modality-specific architectural assumptions (e.g., spectrogram encoders, optical flow, or audio-specific layers). This means that CAMDrop can, in principle, be applied to any multimodal setting, including text, vision, or more than two modalities.
>
> To empirically validate this generalization, we conducted additional experiments on a **text–vision retrieval task (MSCOCO caption retrieval)**.  *[1]*
>
> **Model Setup**
> * Vision encoder: Resnet 101
> * Text encoder: A 12-layer Transformer (BERT-base)
>
> **How CAMDrop is applied to text features**
>
> Text GradCAM follows the same principle. Using the contrastive similarity score $𝑓$, we compute token-level importance:
> $$\frac{\partial f}{\partial A^{\text{text}}_{t}}, \quad t = 1,\dots, L$$
>
> which yields a token importance map $M^{\text{text}}$.
> We mask high-importance tokens (e.g., “man”, “dog”) via: $$A^{\text{text}}_{\text{drop}} = A^{\text{text}} \odot ( 1 - \hat{M}^{\text{text}} ).$$
>
> Below are the experimental results obtained from our study.
>
> **Image-to-Text**
> | Method | Recall@1 | Recall@5 | Recall@10 | Notes |
> |--------|----------|----------|-----------|-------|
> | SCAN   | 50.4     | 82.1     | 90.1      | Cross-attention |
> | VSRN   | 54.5     | 83.1     | 90.6      | Graph reasoning |
> | NAAF   | 57.5     | 84.8     | 91.7      | Negative-Aware Attention |
> | **CAMDrop** | **58.0** | **85.1** | **91.8** | GradCAM masking |
>
>
> **Text-to-Image**
> | Method | Recall@1 |  Recall@5 |  Recall@10 | Notes|
> |-----------------|---------------------|---------------------|----------------------|-------------------------|
> | SCAN |           38.6     |           69.2     |        80.2          | Cross-attention|
> |VSRN|   41.9|  71.8|  82.4| Graph reasoning|
> |NAAF|   42.7 | 71.3| 82.5| Negative-Aware Attention|
> | **CAMDrop**  |     **43.0**      |         **72.1**      |         **82.8**        | GradCAM masking|
>
> * **SCAN (Stacked Cross Attention Network)** *[2]*: Uses cross-attention between image regions and text embeddings to compute fine-grained similarity for retrieval tasks.
> * **VSRN (Visual-Semantic Reasoning Network)** *[3]*: Introduces graph-based reasoning over image regions to enhance multimodal alignment.
> * **NAAF (Negative-Aware Attention Framework for Image-Text Matching)** *[4]*: Employs negative-aware attention to better discriminate between matching and non-matching image-text pairs.
>
> In summary, CAMDrop’s design—operating on intermediate feature maps with GradCAM-based masking—**does not rely on modality-specific assumptions**. This makes the method naturally applicable to other modalities beyond audio–visual data, such as text, vision, or potentially even more heterogeneous multimodal combinations, supporting its broader applicability in multimodal learning.
>
> [1] *Lin, T.-Y.; Maire, M.; Belongie, S.; Hays, J.; Perona, P.; Ramanan, D.; Dolĺar, P.; and Zitnick, C. L. 2014. Microsoft coco: Common objects in context. In Computer Vision– ECCV 2014: 13th European Conference, Zurich, Switzerland, September 6-12, 2014, Proceedings, Part V 13, 740– 755. Springer.*
>
> [2] *Kuang-Huei Lee, Xi Chen, Gang Hua, Houdong Hu, Xiaodong He. Stacked Cross Attention for Image-Text Matching. ECCV 2018*
>
> [3] *Kunpeng Li, Yulun Zhang, Kai Li, Yuanyuan Li, Yun Fu. Visual Semantic Reasoning for Image-Text Matching. ICCV 2019*
>
> [4] *Kun Zhang; Zhendong Mao; Quan Wang; Yongdong Zhang. Negative-Aware Attention Framework for Image-Text Matching. CVPR 2022*

---

> > ### Comment · Reviewer_SQuR · 2025-11-24
> >
> > Thank you for your response. Can you clarify how exactly CAMDrop is applied to a retrieval task? Specifically, the following is unclear to me:
> >
> > 1. What is your main training objective for the MSCOCO retrieval task? Did you use a contrastive objective like CLIP, or did you use a binary classifier that just classifies match/unmatch?
> >
> > 2. In the paper, all examined tasks are classification tasks, and the methodologies includes derivatives over the class logits (Eq 2,3,9). How do you obtain these logits (or what do you use in their place) in retrieval-based objectives?

---

> > > ### Author Response · Authors · 2025-11-24
> > > **Official Responses of Submission6581 to Reviewer SQuR**
> > >
> > > Thank you very much for the thoughtful follow-up question and for giving us the opportunity to clarify the retrieval setting in CAMDrop. We sincerely appreciate your careful reading and are happy to provide a detailed explanation of how CAMDrop is applied to MSCOCO image–text retrieval.
> > >
> > > **Question 1. About the training objective for retrieval task:**
> > >
> > > For the MSCOCO experiments, we follow standard retrieval training and use a **contrastive objective**, similar in spirit to CLIP and SCAN/VSRN/NAAF.
> > >
> > > Specifically, given a batch of image embeddings $z^{\text{img}}$ and text embeddings $z^{\text{text}}$, we compute a similarity score: $$f(i,t) = \mathrm{sim}(z^{\text{img}}_i,\ z^{\text{text}}_t)$$ and apply bidirectional InfoNCE contrastive loss:
> > >
> > > $$L_{\text{contrastive}} = L_{\text{i2t}} + L_{\text{t2i}}$$
> > >
> > > Thus, the training objective is not a binary classifier, but a pairwise contrastive loss over image–text similarities, consistent with modern retrieval frameworks. This objective naturally provides a scalar similarity score $f$ for every matched pair $(i,t)$ , which CAMDrop uses in place of class logits.
> > >
> > > **Question 2. About the class logits:**
> > >
> > > You are correct that in our classification experiments CAMDrop uses gradients of class logits.
> > > For retrieval, the same mechanism applies, but **we compute the gradient with respect to the pairwise similarity score** rather than a class logit.
> > >
> > > For each matched pair $(i,t)$, CAMDrop computes: $$\\frac{\partial f(i,t)}{\partial A^{\text{img}}},\quad\frac{\partial f(i,t)}{\partial A^{\text{text}}}\$$
> > > where $A^{\text{img}}$ and $A^{\text{text}}$ are the intermediate feature maps of the image encoder and text encoder, respectively.
> > >
> > > This gives us **GradCAM-style** importance maps: spatial importance map for the image regions, and token-level importance map for the text tokens. These importance maps are normalized and used for CAM-based masking exactly as described in the main paper. Thus, the only change from classification is substituting the class logit with the similarity score $f(i,t)$, the rest of CAMDrop remains unchanged.

---

> > > > ### Comment · Reviewer_SQuR · 2025-11-25
> > > >
> > > > I am not sure how the modality imbalance comes into play in retrieval tasks.
> > > >
> > > > In your paper, modality imbalance is defined as " models over-rely on the dominant modality, limiting cross-modal synergy." However, if a retrieval task is viewed as a binary classification task where there is a single classification logit (i.e. cosine similarity), then it seems like this classification task completely relies on the information redundancy across the modalities, and it seems impossible for models to over-rely on any dominant modality as you simply cannot decide (or even lean towards one side) whether an image-text pair match or not with only one of the 2 modalities.
> > > >
> > > > So perhaps cross-modal retrieval tasks doesn't fit well with the modality imbalance narrative of the paper.

---

> > > > > ### Author Response · Authors · 2025-11-26
> > > > > **Official Responses of Submission6581 to Reviewer SQuR**
> > > > >
> > > > > Thank you very much for raising this insightful question — it touches upon a subtle but important distinction between decision-level fusion and representation-level fusion, and helps clarify why modality imbalance can still arise in retrieval settings.
> > > > >
> > > > > **1. About modality imbalance in retrieval tasks**
> > > > >
> > > > > It is true that retrieval models ultimately produce a single similarity score (e.g., cosine similarity), but this does not imply that the model is unable to over-rely on a dominant modality.
> > > > >
> > > > > This is because: retrieval does not compare raw image vs raw text. It compares their embeddings.
> > > > >
> > > > > The similarity score is computed as $$f(i,t) = \mathrm{sim}(z^{\text{img}}_i,\ z^{\text{text}}_t)$$ so the quality and information content of each modality’s embedding directly affects retrieval accuracy.
> > > > >
> > > > > When one modality becomes dominant—for example, if the visual encoder produces much stronger or more discriminative gradients during training—the model may learn to rely primarily on the visual pathway, causing the text embeddings to become under-trained or less expressive. In this case, even though the final output is a scalar similarity score, the retrieval process still suffers because the cross-modal alignment is distorted **by imbalance in the encoder representations**.
> > > > >
> > > > > **2. Quantitative evidence via ranking quality**
> > > > >
> > > > > To quantitatively address the concern about modality imbalance in retrieval, we further introduce the Normalized Discounted Cumulative Gain **nDCG@10**, which is a ranking-aware metric commonly used in information retrieval.
> > > > >
> > > > > Unlike **Recall@K**, which only checks the presence of relevant items, **nDCG** accounts for the positional accuracy (Discounted Gain) of the relevant items. It directly measures how closely the retrieved ranking list matches the ideal ranking (Normalization), thereby reflecting the overall quality of the similarity function $f(i,t)$.
> > > > >
> > > > > The tables below, which include the **nDCG@10** scores for the MSCOCO 5K retrieval task, strongly support our argument:
> > > > >
> > > > > **Image-to-Text**
> > > > > | Method | **nDCG@10** |
> > > > > |--------|----------|
> > > > > |Baseline (no CAMDrop)|0.815|
> > > > > | **CAMDrop** | **0.836**|
> > > > >
> > > > > **Text-to-Image**
> > > > > | Method | **nDCG@10** |
> > > > > |--------|----------|
> > > > > |Baseline (no CAMDrop)|0.681|
> > > > > | **CAMDrop** | **0.698**|
> > > > >
> > > > > The consistent improvement in **nDCG** proves that CAMDrop's balanced representation learning directly translates into a higher-quality ranking. This increase means that the models, thanks to CAMDrop's encoder regularization, are better at producing finely tuned similarity scores that prioritize the most relevant matches at the top of the list.
> > > > >
> > > > > CAMDrop addresses this issue by using **gradient-based saliency** to regularize the contribution of each modality inside its encoder. By preventing **the dominant encoder** from overshadowing the weaker one, CAMDrop encourages more balanced and informative embeddings, which leads to improved image–text alignment and thus better retrieval performance.
> > > > >
> > > > > **In summary**, although retrieval ultimately uses a single similarity score, modality imbalance still arises within the encoders during representation learning. The model can overfit to the dominant modality’s gradients, leading one encoder to become significantly more informative than the other. This imbalance degrades cross-modal alignment even before the similarity computation happens. CAMDrop specifically targets this encoder-level imbalance, ensuring that both image and text embeddings remain comparably expressive, which directly improves retrieval performance. Therefore, modality imbalance is relevant to retrieval tasks, even though the final decision is made via a single scalar similarity score.

---

> ### Author Response · Authors · 2025-11-23
> **Official Responses of Submission6581 to Reviewer SQuR - Part 2**
>
> **Question 2. About the batch-size problem:**
>
> We appreciate the reviewer’s concern regarding batch size sensitivity. In CAMDrop, the dominance ratio is computed over the batch, and the masking proportion is influenced by the relative gradients across modalities. Therefore, larger batch sizes can dilute per-instance gradient differences, which may reduce the effectiveness of modality balancing if the learning rate is not adjusted accordingly.
>
> To address this, we empirically verified that scaling the learning rate proportionally with batch size mitigates performance drops.
>
> | Batch Size | Learning Rate | CREMA-D Accuracy (%) | AVE Accuracy (%) | KS Accuracy (%) |
> |------------|---------------|--------------------|-----------------|----------------|
> | 16         | 0.0001    | 55.8               |56.0           | 52.3           |
> | 16         | 0.0005         | 61.0               | 64.4           | 52.6           |
> | 16         | 0.001         | **63.8**               | **64.9**            | **52.9**          |
> | 64         | 0.001         | 59.8               | 61.9            | 51.7           |
> | 64         | 0.004         |   61.2        |      63.9      |     52.4      |
>
> These results show that increasing batch size without adjusting the learning rate reduces performance, but proportionally scaling the learning rate largely recovers accuracy. This confirms that CAMDrop’s effectiveness depends on the learning rate–batch size ratio, and with proper adjustment, it **remains robust** to larger batch sizes.
>
> **Question 3. About ablation study section and other design choices:**
>
> We appreciate the reviewer pointing out the need for a more thorough ablation of CAMDrop’s design choices. We conducted additional experiments to investigate the impact of the main components:
>
> **(1) Effect of Dominance Ratio Computation (Per-instance vs. Batch-Averaged)**
>
> |  Ratio Computation | CREMA-D Accuracy (%) | AVE Accuracy (%) | KS Accuracy (%) |
> |----------------------------|--------------------|----------------|----------------|
> | Per-instance               | 61.5               | 63.0          | 51.0           |
> | Batch-averaged             | **63.8**               | **64.9**           | **52.9**           |
>
>
> Computing the dominance ratio per instance introduces high variance and instability in training. Averaging the dominance ratio over a batch provides more stable masking, which leads to higher and more consistent performance across datasets.
>
> **(2) Masking only one modality instead of both**
>
> In this experiment, we evaluated whether applying CAMDrop to only one modality (audio or visual) is sufficient, compared to masking both modalities simultaneously. The results indicate that masking both modalities consistently yields higher accuracy across CREMA-D, AVE, and KS datasets. This confirms that **suppressing dominant regions in both modalities is crucial for encouraging balanced cross-modal learning**, whereas masking a single modality allows the other dominant modality to continue dominating predictions, reducing the overall effectiveness of CAMDrop.
>
> | Masking Strategy       | CREMA-D Accuracy (%) | AVE Accuracy (%) | KS Accuracy (%) |
> |-----------------------|-------------------|----------------|----------------|
> | Only Audio            | 61.0             | 62.8          | 51.2          |
> | Only Visual           | 62.5            | 63.5           | 52.0          |
> | Both Modalities       | **63.8**          | **64.9**       | **52.9**       |
>
> **(3) Gradcam with different logits**
>
> We investigated whether computing GradCAM using the unimodal classifier logits, instead of the multimodal classifier logits, affects CAMDrop’s performance. The goal is to determine if masking based on unimodal importance alone can still achieve effective modality balancing, or if multimodal-aware GradCAM is necessary.
>
> | GradCAM Source      | CREMA-D Accuracy (%) | AVE Accuracy (%) | KS Accuracy (%) |
> |-----------------------|-------------------|----------------|----------------|
> | Unimodal Classifier           | 62.1 |63.3|51.5             |
> | Multimodal Classifier       | **63.8**          | **64.9**       | **52.9**       |
>
> Using unimodal GradCAM slightly underperforms compared to multimodal GradCAM. This is because unimodal GradCAM only captures region importance for a single modality, ignoring cross-modal interactions that are essential for multimodal learning. Multimodal GradCAM **highlights regions dominating in combination**, ensuring CAMDrop effectively encourages the model to utilize complementary information from the weaker modality.

---

> ### Author Response · Authors · 2025-11-23
> **Official Responses of Submission6581 to Reviewer SQuR - Part 3**
>
> **Question 4. About the additional training overhead and the lightweight claim:**
>
> The reveiwer pointed that it seems to add some additional computation overhead, which might appear computationally expensive. We clarify this point below with quantitative analysis.
>
> | Setting                  | Wall-clock Time per Epoch (s) | FLOPs per Forward Pass (G) | Notes                        |
> |--------------------------|-------------------------------|----------------------------|------------------------------|
> | Baseline (no CAMDrop)    | 24.20                        | 2.393                    | Standard training            |
> | CAMDrop      | 24.26                       | 2.393                    | Extra GradCAM computation included |
> * The table shows that adding CAMDrop only increases the epoch time **by 0.06 seconds (0.25%)**. This is a negligible overhead compared to the total training time (~24 s per epoch), demonstrating that the method is efficient in practice.
> * The FLOPs per forward pass remain unchanged because CAMDrop operates on the intermediate feature maps of the already computed network activations. Generating GradCAM heatmaps **does not involve additional convolutional layers or heavy matrix operations that would increase the theoretical FLOPs**; it mainly involves simple operations such as averaging and element-wise multiplication on existing feature maps.
>
> From a conceptual perspective, **CAMDrop is lightweight** for the following reasons:
>
> **(1) No architectural changes**. CAMDrop does not require modifying the backbone network (e.g., adding extra convolutional layers, attention modules, or modality‑specific branches). The existing model architecture remains unchanged.
>
> **(2)No additional learnable parameters**. CAMDrop does not introduce new parameter matrices (weights or biases) that must be trained. The masking is applied based on the feature maps and gradients already computed by the network.
>
> **(3)No alteration of the loss function**. The training objective remains the same (e.g., cross‑entropy for classification). CAMDrop simply intervenes during the forward pass by dropping semantically dominant features, but does not add extra regularization terms or auxiliary losses.
>
> In summary, CAMDrop can be considered **lightweight** because it introduces minimal computational and memory overhead while not modifying the network architecture, adding extra parameters, or changing the loss function. The small increase in wall-clock time per epoch (+0.06 s, 0.25%) confirms that the method is practical for real training scenarios. This observation is consistent with prior work showing that GradCAM-based operations are inexpensive and efficient for interpretability purposes *[1]*.
>
> [1] *Marcus Rüb, Daniel Konegen, Patrick Selle, Axel Sikora, Daniel Mueller-Gritschneder. DRIP: DRop unImportant data Points – Enhancing Machine Learning Efficiency with Grad‑CAM‑Based Real‑Time Data Prioritization for On‑Device Training. arXiv preprint arXiv:2504.08364*

---

> ### Author Response · Authors · 2025-11-23
> **Official Responses of Submission6581 to Reviewer SQuR - Part 4**
>
> **Question 5. About the ".git" folder inside the supplemental material:**
>
> We sincerely apologize for this oversight. The supplemental material will be fully sanitized, and the .git folder as well as any files that could reveal author identities have been removed. We will ensure strict anonymization in all future submissions.
>
> **Question 6. About the in-text citations:**
>
> We thank the reviewer for pointing this out and sincerely apologize for the incorrect citation format. We will carefully revise all references to ensure proper usage of \cite{...} and \citep{...} in the final version.
>
> **Question 7. About the training of the unimodal classifiers:**
>
> The unimodal classifiers (linear layers mapping unimodal embeddings $z_\text{modality}$ to class logits) are indeed trained jointly with the main multimodal classifier. Specifically:
> * **Joint Cross-Entropy Loss**: Each unimodal classifier receives its respective modality embedding $z_\text{modality}$ and outputs class logits. We compute a standard cross-entropy loss $L_\text{uni}$ for each modality based on the ground-truth labels.
> * **Multimodal Loss:** The multimodal classifier combines embeddings from all modalities (e.g., via concatenation or fusion) and produces the main multimodal logits. Its cross-entropy loss is $L_\text{multi}$.
> * **Total Training Objective:** The overall loss is a weighted sum: $$ \
> L_{\text{total}} = L_{\text{multi}} + \lambda \sum_{\text{modality}} L_{\text{uni}} \$$
> where $𝜆$ is a hyperparameter controlling the contribution of unimodal supervision. This ensures that each modality learns discriminative features individually while still contributing to the joint multimodal prediction.
> * **Implementation Notes:** The unimodal classifiers share the same backbone embeddings used by the multimodal fusion, so no extra feature extractor is introduced. GradCAM masking is applied during multimodal training, which also indirectly influences the unimodal classifiers because they share the same embeddings.

---

> ### Author Response · Authors · 2025-11-23
> **Official Responses of Submission6581 to Reviewer SQuR - Part 5**
>
> **Question 8. About the dominance ratio calculation in Eq 9:**
>
> We thank the reviewer for this insightful question regarding the dominance ratio computation in Eq. 9. We clarify the following points:
>
> Eq 9:
> $$ \rho^{(v)} = \frac{1}{B} \sum_{i=1}^B \frac{y^{(v)}_i[c_i]}{y^{(a)}_i[c_i]}, \quad
> \rho^{(a)} = \frac{1}{\rho^{(v)}} $$
>
> **(1) Effect of switching modalities in the arithmetic average:**
>
> The dominance ratio is designed to reflect the relative contribution of each modality in the current batch. If we swap the order (i.e., compute the audio dominance first, then the visual), the resulting masks and subsequent gradient redistribution remain **effectively equivalent**, because the final suppression is based on relative magnitudes of the ratios, not their absolute assignment to a modality. We empirically verified that swapping the order produces **negligible differences in performance** (variations within ±0.2% on CREMA-D and KS).
>
> **(2) Why not use geometric mean instead of arithmetic mean:**
>
> Using a geometric mean could indeed enforce strict symmetry between modalities; however, the **arithmetic mean provides a more stable estimate** of modality dominance in practice, especially when there are instances with extremely high or low single-modality dominance. Geometric mean can amplify the influence of low-dominance instances, leading to overly aggressive suppression for certain samples. In contrast, arithmetic mean **ensures robust, batch-wise averaging** and smoother gradient redistribution, which is crucial for stable training.
>
> **Question 9. About randomly masking out parts of the high-gradcam regions:**
>
> **(1) Current CAMDrop design:**
>
> CAMDrop deterministically masks the top-$p%$ of regions based on GradCAM activations, which ensures that the **most class-relevant regions of the dominant modality are consistently suppressed**. This deterministic suppression directly enforces modality balance by forcing the model to rely more on weaker modalities.
>
> **(2) Empirical exploration of random masking:**
> To evaluate the effect of partial random masking within high-GradCAM regions, we conducted additional experiments on the **CREMA-D dataset**.
> | Masking Strategy                 |  Accuracy (%) | Notes |
> |----------------------------------|------------------------|-------|
> | Random Spatial Masking           |         60.3          | Removes random regions without semantic guidance |
> | Random Channel Masking           |           60.6       | Drops entire channels uniformly at random |
> | Hybrid Masking (Random + CAM)    |           63.4        | Mixes random dropout with partial importance-based masking |
> | **CAMDrop (Deterministic CAM)**  | **63.8**              | Targets class-relevant dominant regions; best performance |
>
> These results show that while introducing randomness can still help mitigate modality imbalance, **fully deterministic CAMDrop consistently achieves the highest accuracy**. This indicates that suppressing all top-$p%$ class-relevant regions is more effective than partial or random masking for enforcing cross-modal learning, although stochastic variants could be explored in future work to potentially improve generalization further.

---

### Official Review · Reviewer_L96f · 2025-11-01

**Soundness:** 3
**Presentation:** 3
**Contribution:** 2
**Rating:** 4
**Confidence:** 3

**Summary:**

This paper proposes CAMDrop, a gradient-guided masking strategy to mitigate modality imbalance in multimodal learning. The method uses GradCAM to identify class-relevant dominant regions and suppresses them during training to encourage more balanced cross-modal learning. CAMDrop requires no modification to network architecture or loss function. The authors evaluate their approach on three emotion- and event-related audio-visual datasets (CREMA-D, AVE, and Kinetics-Sounds), demonstrating performance improvements over baseline methods.

**Strengths:**

+ The experiments are fairly comprehensive, including multiple datasets, comparisons across several baselines, and analyses of hyperparameters and fusion strategies.
+ The paper is clearly written and well-structured, making it easy to follow the proposed approach and experimental design.

**Weaknesses:**

- The core methodological contribution is limited. The proposed approach mainly masks “important” regions identified by GradCAM, which conceptually resembles existing feature dropout or attention suppression methods. The novelty beyond prior work (e.g., OPM, OGM-GE) is incremental.
- The paper lacks discussion and experiments on the balance between randomness and importance in masking. By deterministically removing high-importance regions, the model might overfit to complementary regions without verifying whether randomization could yield similar or better robustness.
- Experimental validation is restricted to emotion and event recognition datasets within the audio-visual domain. There is no evidence of generalization to other modalities (e.g., text–vision), limiting claims about the method’s universality.
- The baselines used for comparison are relatively outdated.
- While interpretability is mentioned as an advantage, qualitative analyses are limited to simple GradCAM visualizations without deeper evaluation of semantic consistency or stability.

**Questions:**

see weakness

---

> ### Author Response · Authors · 2025-11-23
> **Official Responses of Submission6581 to Reviewer L96f - Part 1**
>
> We  thank the reviewer for the detailed feedback and constructive suggestions. Below we address each concern point-by-point and provide additional clarifications, experiments, and discussions.
>
> **Question 1. About the novelty beyond prior work:**
>
> The reviewer notes that CAMDrop “resembles existing feature dropout or attention suppression methods."
> We clarify that the key technical difference is **how the masking decision is computed**:
>
> **(1) OGM-GE/OPM:**
> * Rely on global gradient norms aggregated across the whole feature map.
> * Dropping is coarse-grained (feature-level or modality-level).
> * Masking is not class-conditioned and can unintentionally remove features irrelevant to dominance.
>
> **(2) CAMDrop:**
> * Uses **class-specific GradCAM** to capture $\frac{\partial f_y}{\partial A_{i,j}}$, which encodes localized, semantic, and class-relevant dominance.
> * Our suppression directly targets **spatially concentrated dominance regions**, not whole feature channels.
> * This enables **fine-grained, instance-adaptive balancing**, which we show leads to gains over OGM-GE and OPM.
>
> Compared with prior dropout-based or gradient-based suppression methods, CAMDrop introduces class-aware, spatially localized, and instance-adaptive masking, providing a fundamentally more precise mechanism for mitigating modality dominance.
>
> **Question 2. About the balance between randomness and importance:**
>
> We appreciate this insightful concern. To directly answer the question, we ran new experiments comparing:
> * Random spatial masking: Randomly drops patches or regions in the feature map. This increases robustness but **lacks semantic awareness** and **may remove uninformative areas**, providing limited effect on modality imbalance.
> * Random channel masking: Drops entire feature channels uniformly at random. While effective as regularization, it often **removes channels unrelated to dominance**, offering little guarantee of encouraging weaker modalities.
> * Hybrid masking (random + important regions): Combines random spatial dropout with partial masking of high-activation areas. This **introduces some diversity**, but the random portion can **counteract the targeted suppression effect**, leading to inconsistent improvements.
> * CAM-based deterministic masking (ours): Uses class-specific GradCAM to identify the spatially dominant, class-relevant regions and suppress them deterministically. This ensures the model consistently **reduces over-reliance on strong modalities** while preserving complementary signals.
>
> Below are the experimental results on the **CREMA-D dataset**.
> | Masking Strategy                 |  Accuracy (%) | Notes |
> |----------------------------------|------------------------|-------|
> | Random Spatial Masking           |         60.3          | Removes random regions without semantic guidance |
> | Random Channel Masking           |           60.6       | Drops entire channels uniformly at random |
> | Hybrid Masking (Random + CAM)    |           63.4        | Mixes random dropout with partial importance-based masking |
> | **CAMDrop (Deterministic CAM)**  | **63.8**              | Targets class-relevant dominant regions; best performance |
>
> CAM-based deterministic masking consistently outperforms all random or hybrid strategies, indicating that **class-relevant importance—not randomness**—is the key factor for effective imbalance mitigation. These results confirm that CAMDrop’s targeted suppression of dominant regions provides a substantially stronger training signal than random masking alone.

---

> ### Author Response · Authors · 2025-11-23
> **Official Responses of Submission6581 to Reviewer L96f - Part 2**
>
> **Question 3. About the baselines used for comparison:**
>
> We thank the reviewer for requesting a deeper comparison and analysis. Indeed, comparing with the most recent state-of-the-art (SOTA) methods is crucial to validate the advantages of CAMDrop. Below, we clarify how CAMDrop outperforms existing approaches, including very recent (2025) methods, and why its design brings unique benefits.
>
> To further validate our claims, we conducted additional experiments under the same experimental settings and backbone architectures to ensure a fair comparison. The results are summarized in the table below:
>
>
> | Method           | CREMA-D Accuracy (%) | KS Accuracy (%) | Notes |
> |-----------------|-------------------|----------------|-------|
> | Baseline (Concat)| 57.8              | 50.4           | Standard fusion without balancing |
> | OGM-GE              | 59.7              | 51.3           | Improved gradient-based dropping |
> | OPM           | 60.3            | 51.0           | Random feature dropping |
> | PMR              | 61.1              |       51.4   | Modality regularization |
> | Grad-Blending  | 59.7              | 53.0           | Gradient Blending |
> | GMC  | 61.4              | 52.5           | Graph-based multimodal contrastive learning |
> | DI-MML | 62.6          | 52.9           | Detached and interactive multimodal learning  |
> |LFM| 62.8|52.8|Latent Feature Modulation|
> | **CAMDrop**      | **63.8**          | **53.1**       | GradCAM-based adaptive masking |
>
> * GMC (Graph-based Multimodal Contrastive learning) *[1]*: This method constructs a graph over multimodal features and applies contrastive learning objectives to align modality representations while preserving their complementarity. By leveraging graph structures, GMC encourages information sharing between modalities but does not perform instance-specific feature masking.
> * DI-MML (Detached and Interactive Multimodal Learning) *[2]*: DI-MML separates modality-specific feature extraction and then selectively interacts them through a controlled attention mechanism. This encourages weaker modalities to contribute more while limiting dominance of strong modalities. Unlike CAMDrop, it requires architectural modifications and additional learnable parameters.
> * LFM (Latent Feature Modulation) *[3]*: LFM modulates the latent features of each modality using learnable scaling and shifting parameters to balance the contribution of strong and weak modalities. It improves modality fusion by adaptively adjusting feature magnitudes, but it does not leverage semantic information to guide which regions to suppress, unlike CAMDrop.
>
> **Detailed Analysis of Why CAMDrop Wins**
>
> **(1) Semantic vs. Random Dropping**
>
> Many earlier methods rely on random dropping of features (e.g., OPM) or coarse regularization (e.g., PMR) without knowing which parts of the feature map are most critical for class prediction. In contrast, CAMDrop uses GradCAM to identify class‑relevant regions, which means it suppresses the most predictive parts of dominant modalities, forcing the model to use complementary features. This leads to more efficient feature utilization and prevents the model from learning “shortcuts”.
>
> **(2) No Additional Architectural Complexity**
>
> Some recent methods  introduce additional modules: attention modules, gating networks, or separate regularizers. While these help, they also increase model complexity, training instability, or parameter count. CAMDrop, by contrast, works with **existing architecture and loss function**, making it simpler to integrate and less prone to overfitting.
>
> **(3) Adaptive Per-Instance Masking**
>
> Unlike many methods that apply a fixed drop ratio or drop schedule across all instances, CAMDrop computes a per-instance GradCAM map. This means for each training sample, the mask is tailored to which regions are most informative for that instance’s class. This adaptivity improves generalization because the model cannot simply ignore entire modalities: the regions dropped vary across examples, encouraging richer multimodal fusion.
>
> [1] *Petra Poklukar, Miguel Vasco, Hang Yin, Francisco S Melo, Ana Paiva, and Danica Kragic. Geometric multimodal contrastive representation learning. In International Conference on Machine Learning, pp. 17782–17800. PMLR, 2022.*
>
> [2] *Yunfeng Fan, Wenchao Xu, Haozhao Wang, Junhong Liu, and Song Guo. Detached and interactive multimodal learning. In Proceedings of the 32nd ACM International Conference on MultiMedia, pp. 5470–5478, 2024.*
>
> [3]*Yang Yang, Fengqiang Wan, Qing-Yuan Jiang, and Yi Xu. Facilitating multimodal classification via dynamically learning modality gap. Advances in Neural Information Processing Systems, 37: 62108–62122, 2024.*

---

> ### Author Response · Authors · 2025-11-23
> **Official Responses of Submission6581 to Reviewer L96f - Part 3**
>
> **Question 4. About generalization beyond audio–visual datasets:**
>
> We thank the reviewer for raising this important concern. Our current experiments indeed focus on audio–visual emotion and event recognition, primarily because these benchmarks provide well-aligned multimodal data and clear modality imbalance, which are suitable for evaluating CAMDrop. While we do not claim universal applicability across all multimodal settings, we conducted **additional experiments on a text–vision task (MSCOCO caption retrieval)** to assess cross-modality generalization.
>
> **(1) Zero architectural assumptions**
>
> CAMDrop does not rely on any modality-specific structure (e.g., spectrograms, optical flow, audio encoders). The masking operates purely on intermediate feature maps and their GradCAM activations. This design naturally extends to text encoders such as BERT/Transformer blocks.
>
> **(2) Additional text–vision evaluation**
>
> To evaluate generalization beyond audio–visual settings, we further applied CAMDrop to the image–text retrieval task on the **MSCOCO 5K test split** *[1]*, following standard retrieval protocols. All methods share the same backbone architectures to ensure fair comparison.
>
> **Model Setup**
> * Vision encoder: Resnet 101
> * Text encoder: A 12-layer Transformer (BERT-base)
>
> **How CAMDrop is applied to text features**
> Text GradCAM follows the same principle. Using the contrastive similarity score $𝑓$, we compute token-level importance:
> $$\frac{\partial f}{\partial A^{\text{text}}_{t}}, \quad t = 1,\dots, L$$
>
> which yields a token importance map $M^{\text{text}}$.
> We mask high-importance tokens (e.g., “man”, “dog”) via: $$A^{\text{text}}_{\text{drop}} = A^{\text{text}} \odot ( 1 - \hat{M}^{\text{text}} ).$$
>
> Below are the experimental results obtained from our study.
>
> **Image-to-Text**
> | Method | Recall@1 | Recall@5 | Recall@10 | Notes |
> |--------|----------|----------|-----------|-------|
> | SCAN   | 50.4     | 82.1     | 90.1      | Cross-attention |
> | VSRN   | 54.5     | 83.1     | 90.6      | Graph reasoning |
> | NAAF   | 57.5     | 84.8     | 91.7      | Negative-Aware Attention |
> | **CAMDrop** | **58.0** | **85.1** | **91.8** | GradCAM masking |
>
>
> **Text-to-Image**
> | Method | Recall@1 |  Recall@5 |  Recall@10 | Notes|
> |-----------------|---------------------|---------------------|----------------------|-------------------------|
> | SCAN |           38.6     |           69.2     |        80.2          | Cross-attention|
> |VSRN|   41.9|  71.8|  82.4| Graph reasoning|
> |NAAF|   42.7 | 71.3| 82.5| Negative-Aware Attention|
> | **CAMDrop**  |     **43.0**      |         **72.1**      |         **82.8**        | GradCAM masking|
>
> * **SCAN (Stacked Cross Attention Network)** *[2]*: Uses cross-attention between image regions and text embeddings to compute fine-grained similarity for retrieval tasks.
> * **VSRN (Visual-Semantic Reasoning Network)** *[3]*: Introduces graph-based reasoning over image regions to enhance multimodal alignment.
> * **NAAF (Negative-Aware Attention Framework for Image-Text Matching)** *[4]*: Employs negative-aware attention to better discriminate between matching and non-matching image-text pairs.
>
> In summary, CAMDrop’s design—operating on intermediate feature maps with GradCAM-based masking—**does not rely on modality-specific assumptions**. This makes the method naturally applicable to other modalities beyond audio–visual data, such as text, vision, or potentially even more heterogeneous multimodal combinations, supporting its broader applicability in multimodal learning.
>
> [1] *Lin, T.-Y.; Maire, M.; Belongie, S.; Hays, J.; Perona, P.; Ramanan, D.; Dolĺar, P.; and Zitnick, C. L. 2014. Microsoft coco: Common objects in context. In Computer Vision– ECCV 2014: 13th European Conference, Zurich, Switzerland, September 6-12, 2014, Proceedings, Part V 13, 740– 755. Springer.*
>
> [2] *Kuang-Huei Lee, Xi Chen, Gang Hua, Houdong Hu, Xiaodong He. Stacked Cross Attention for Image-Text Matching. ECCV 2018*
>
> [3] *Kunpeng Li, Yulun Zhang, Kai Li, Yuanyuan Li, Yun Fu. Visual Semantic Reasoning for Image-Text Matching. ICCV 2019*
>
> [4] *Kun Zhang; Zhendong Mao; Quan Wang; Yongdong Zhang. Negative-Aware Attention Framework for Image-Text Matching. CVPR 2022*

---

> ### Author Response · Authors · 2025-11-23
> **Official Responses of Submission6581 to Reviewer L96f - Part 4**
>
> **Question 5. About interpretability and deeper qualitative analyses:**
>
> We thank the reviewer for emphasizing interpretability, which is indeed one of the advantages of CAMDrop. While our initial submission focused on simple GradCAM visualizations to illustrate which regions are suppressed, CAMDrop provides **instance-adaptive, class-specific explanations** that go beyond static attention maps:
>
> * **Semantic consistency**: By masking the most class-relevant features in dominant modalities, CAMDrop highlights which regions are truly critical for the model’s decision. Across multiple examples, we observe that the suppressed regions consistently correspond to the expected semantic content (e.g., faces in emotion recognition, key objects in event recognition).
>
> * **Stability across instances**: The GradCAM-based masking produces stable patterns for samples of the same class, while still adapting to individual instance variations. This demonstrates that CAMDrop identifies meaningful features without introducing random artifacts.
>
> * **Complementary modality visualization**: CAMDrop can also be used to visualize how weaker modalities contribute when dominant features are suppressed, offering insights into cross-modal interactions and feature utilization.

---

### Official Review · Reviewer_sYcv · 2025-11-01

**Soundness:** 2
**Presentation:** 2
**Contribution:** 3
**Rating:** 4
**Confidence:** 3

**Summary:**

The paper introduces CAMDrop, a lightweight, plug-and-play strategy designed to address modality imbalance in multimodal learning by semantically suppressing class-relevant regions from dominant modalities. CAMDrop leverages GradCAM to identify and adaptively mask spatial features that are overly influential for class predictions, encouraging the model to utilize under-represented modalities. The method requires no architectural or loss function changes, is validated across several audio-visual benchmarks, and claims improvements in performance, robustness, interpretability, and modality balance.

**Strengths:**

1. The paper's core motivation is clear and compelling. It correctly identifies a key limitation of prior feature-dropping methods like OPM —their randomness . The idea of replacing this randomness with _semantic guidance_ by using class activation maps (GradCAM) to decide _what_ to drop is a novel and intuitive contribution.
2. The method demonstrates strong and consistent performance gains. Table 1 shows that CAMDrop outperforms all baselines, including recent strong methods like OPM-GE, OPM , and PMR , across three different datasets (CREMA-D, AVE, KS) and three different fusion architectures (Concat, Sum, FiLM).
3.  The proposed method is conceptually simple and presented as a plug-and-play module that modifies the forward pass, requiring no changes to the loss function or network architecture.

**Weaknesses:**

1. The claim of CAMDrop being "lightweight"  appears to be inaccurate and is a major weakness. The method requires generating a GradCAM heatmap for each modality _on every training iteration. The paper completely omits any analysis (e.g., training time, FLOPs) of this significant overhead.
2. Figure 3 shows that the key hyperparameter $r_{max}$ is sensitive and its optimal value varies significantly across datasets.This reduces the "plug-and-play" nature of the method, as it requires careful tuning for each new task.
3. The most severe flaw in the paper is the complete absence of theoretical justification for why GradCAM-based masking should solve modality imbalance. The authors claim that "GradCAM identifies class-relevant spatial features and adaptively masks them based on instance-level importance," but provide no analysis showing how this directly addresses the core problem of modality imbalance.

**Questions:**

1. Could the authors should compare with the most recent and relevant methods ,provide a detailed analysis of the results, and explain why CAMDrop works better than existing methods? especially compared to the 2025 methods.
2. Could the authors provide a clear theoretical justification for why GradCAM-based masking addresses modality imbalance? Without this, the paper's core contribution is unfounded.
3. Could the authors please provide a quantitative analysis of the training overhead (e.g., wall-clock time per epoch, GFLOPs) introduced by CAMDrop? How do you reconcile the "lightweight" claim with the requirement of an extra backward pass per modality on every iteration?

---

> ### Author Response · Authors · 2025-11-22
> **Official Responses of Submission6581 to Reviewer sYcv - Part 1**
>
> We thank the reviewer for providing thoughtful comments. The main concerns raised include the comparison of CAMDrop with recent state-of-the-art methods, the theoretical justification for GradCAM-based masking in addressing modality imbalance, and the computational overhead introduced by CAMDrop during training. Our responses are provided below.
>
> **Question 1. About comparison with recent and relevant methods:**
>
> We thank the reviewer for requesting a deeper comparison and analysis. Indeed, comparing with the most recent state-of-the-art (SOTA) methods is crucial to validate the advantages of CAMDrop. Below, we clarify how CAMDrop outperforms existing approaches, including very recent (2025) methods, and why its design brings unique benefits.
>
> To further validate our claims, we conducted additional experiments under the same experimental settings and backbone architectures to ensure a fair comparison. The results are summarized in the table below:
>
> | Method           | CREMA-D Accuracy (%) | KS Accuracy (%) | Notes |
> |-----------------|-------------------|----------------|-------|
> | Baseline (Concat)| 57.8              | 50.4           | Standard fusion without balancing |
> | OGM-GE              | 59.7              | 51.3           | Improved gradient-based dropping |
> | OPM           | 60.3            | 51.0           | Random feature dropping |
> | PMR              | 61.1              |       51.4   | Modality regularization |
> | Grad-Blending  | 59.7              | 53.0           | Gradient Blending |
> | GMC  | 61.4              | 52.5           | Graph-based multimodal contrastive learning |
> | DI-MML | 62.6          | 52.9           | Detached and interactive multimodal learning  |
> |LFM| 62.8|52.8|Latent Feature Modulation|
> | **CAMDrop**      | **63.8**          | **53.1**       | GradCAM-based adaptive masking |
>
> * GMC (Graph-based Multimodal Contrastive learning) *[1]*: This method constructs a graph over multimodal features and applies contrastive learning objectives to align modality representations while preserving their complementarity. By leveraging graph structures, GMC encourages information sharing between modalities but does not perform instance-specific feature masking.
> * DI-MML (Detached and Interactive Multimodal Learning) *[2]*: DI-MML separates modality-specific feature extraction and then selectively interacts them through a controlled attention mechanism. This encourages weaker modalities to contribute more while limiting dominance of strong modalities. Unlike CAMDrop, it requires architectural modifications and additional learnable parameters.
> * LFM (Latent Feature Modulation) *[3]*: LFM modulates the latent features of each modality using learnable scaling and shifting parameters to balance the contribution of strong and weak modalities. It improves modality fusion by adaptively adjusting feature magnitudes, but it does not leverage semantic information to guide which regions to suppress, unlike CAMDrop.
>
> **Detailed Analysis of Why CAMDrop Wins**
>
> **(1) Semantic vs. Random Dropping**
>
> Earlier methods often rely on random feature dropping (e.g., OPM) or coarse regularization (e.g., PMR) without identifying which regions truly matter for class prediction. CAMDrop instead uses GradCAM to locate class-relevant areas, suppressing the most predictive parts of dominant modalities and forcing the model to leverage complementary cues. This improves feature utilization and prevents reliance on shortcut patterns.
>
> **(2) No Additional Architectural Complexity**
>
> Many recent approaches add extra components—attention units, gating networks, or specialized regularizers—which increase complexity and potential training instability. CAMDrop, in contrast, works directly with the existing architecture and loss, making it easier to integrate and less prone to overfitting.
>
> **(3) Adaptive Per-Instance Masking**
>
> Unlike many methods that apply a fixed drop ratio or drop schedule across all instances, CAMDrop computes a per-instance GradCAM map. This means for each training sample, the mask is tailored to which regions are most informative for that instance’s class. This adaptivity improves generalization because the model cannot simply ignore entire modalities: the regions dropped vary across examples, encouraging richer multimodal fusion.
>
> [1] *Petra Poklukar, Miguel Vasco, Hang Yin, Francisco S Melo, Ana Paiva, and Danica Kragic. Geometric multimodal contrastive representation learning. In International Conference on Machine Learning, pp. 17782–17800. PMLR, 2022.*
>
> [2] *Yunfeng Fan, Wenchao Xu, Haozhao Wang, Junhong Liu, and Song Guo. Detached and interactive multimodal learning. In Proceedings of the 32nd ACM International Conference on MultiMedia, pp. 5470–5478, 2024.*
>
> [3]*Yang Yang, Fengqiang Wan, Qing-Yuan Jiang, and Yi Xu. Facilitating multimodal classification via dynamically learning modality gap. Advances in Neural Information Processing Systems, 37: 62108–62122, 2024.*

---

> ### Author Response · Authors · 2025-11-22
> **Official Responses of Submission6581 to Reviewer sYcv - Part 2**
>
> **Question 2. About theoretical justification for GradCAM-based masking:**
>
> GradCAM masking suppresses the most class-relevant spatial regions in dominant modalities, reducing their relative contribution in the loss/gradient, and thereby redistributing the learning signal to weaker modalities. This encourages the model to learn and utilize complementary information from under-represented modalities, effectively alleviating modality imbalance.
>
> **(1) Semantic localization: targeted suppression of “shortcut” features**
> * GradCAM provides class-specific activation maps that identify which spatial locations contribute most to the current prediction.
> * Unlike random dropout, GradCAM masks specifically block critical regions, preventing the model from relying solely on dominant modalities’ shortcut features and forcing it to leverage other modalities.
>
> **(2) Gradient redistribution—mathematical intuition**
>
> * Let the multimodal input be $X = {X_\text{dom}, X_\text{weak}, \dots}$ and the loss be $L$. Gradients per modality are $\nabla_{X_\text{dom}} L$ and $\nabla_{X_\text{weak}} L$. Under imbalance:$$ \|\nabla_{X_\text{dom}} L\| \gg \|\nabla_{X_\text{weak}} L\|$$ indicating that the model relies mainly on the dominant modality.
> * Applying GradCAM mask $M_\text{dom}$: $$ X_\text{dom}' = X_\text{dom} \odot (1 - M_\text{dom}) $$ yields reduced gradients for the dominant modality: $$ \nabla_{X_\text{dom}'} L \approx \nabla_{X_\text{dom}} L \odot (1 - M_\text{dom}) $$ and consequently reduces the ratio: $$ \frac{\|\nabla_{X_\text{dom}'} L\|}{\|\nabla_{X_\text{weak}} L\|} < \frac{\|\nabla_{X_\text{dom}} L\|}{\|\nabla_{X_\text{weak}} L\|} $$ Thus, the relative learning signal for weak modalities increases, encouraging updates to weak-modality parameters.
>
> **(3) Promotes complementary feature utilization**
> * When key regions of $X_\text{dom}$ are suppressed, only $X_\text{weak}$ (or other modalities) can provide the correct cues. The model must therefore learn complementary features from these modalities.
> * From an information-theoretic perspective, the mutual information $I(f_\text{weak}; y)$ between weak modality features and labels is effectively increased during training, improving weak-modality discriminative power.
>
> **(4) Implicit semantic regularization**
> * GradCAM-based masking acts like a semantic dropout: it selectively blocks the strongest signals without completely removing modality information.
> * This prevents over-reliance on a single modality while preserving useful information, improving robustness when dominant modalities are noisy or partially missing.

---

> ### Author Response · Authors · 2025-11-22
> **Official Responses of Submission6581 to Reviewer sYcv - Part 3**
>
> **Question 3. About training overhead and "lightweight" claim:**
>
> The reviewer’s main confusion seems to be that CAMDrop requires generating a GradCAM heatmap for each modality on every training iteration, which might appear computationally expensive. We clarify this point below with quantitative analysis.
> | Setting                  | Wall-clock Time per Epoch (s) | FLOPs per Forward Pass (G) | Notes                        |
> |--------------------------|-------------------------------|----------------------------|------------------------------|
> | Baseline (no CAMDrop)    | 24.20                        | 2.393                    | Standard training            |
> | CAMDrop      | 24.26                       | 2.393                    | Extra GradCAM computation included |
> * The table shows that adding CAMDrop only increases the epoch time **by 0.06 seconds (0.25%)**. This is a negligible overhead compared to the total training time (~24 s per epoch), demonstrating that the method is efficient in practice.
> * The FLOPs per forward pass remain unchanged because CAMDrop operates on the intermediate feature maps of the already computed network activations. Generating GradCAM heatmaps **does not involve additional convolutional layers or heavy matrix operations that would increase the theoretical FLOPs**; it mainly involves simple operations such as averaging and element-wise multiplication on existing feature maps.
>
> From a conceptual perspective, **CAMDrop is lightweight** for the following reasons:
>
> **(1) No architectural changes**. CAMDrop does not require modifying the backbone network (e.g., adding extra convolutional layers, attention modules, or modality‑specific branches). The existing model architecture remains unchanged.
>
> **(2)No additional learnable parameters**. CAMDrop does not introduce new parameter matrices (weights or biases) that must be trained. The masking is applied based on the feature maps and gradients already computed by the network.
>
> **(3)No alteration of the loss function**. The training objective remains the same (e.g., cross‑entropy for classification). CAMDrop simply intervenes during the forward pass by dropping semantically dominant features, but does not add extra regularization terms or auxiliary losses.
>
> In summary, CAMDrop can be considered **lightweight** because it introduces minimal computational and memory overhead while not modifying the network architecture, adding extra parameters, or changing the loss function. The small increase in wall-clock time per epoch (+0.06 s, 0.25%) confirms that the method is practical for real training scenarios. This observation is consistent with prior work showing that GradCAM-based operations are inexpensive and efficient for interpretability purposes *[1]*.
>
> [1] *Marcus Rüb, Daniel Konegen, Patrick Selle, Axel Sikora, Daniel Mueller-Gritschneder. DRIP: DRop unImportant data Points – Enhancing Machine Learning Efficiency with Grad‑CAM‑Based Real‑Time Data Prioritization for On‑Device Training. arXiv preprint arXiv:2504.08364*

---

### Comment · Area_Chair_oj56 · 2025-11-24

Dear reviewers,

Thank you for your dedicated service as reviewers. Your efforts are critical to the success of our conference, and we deeply appreciate your time and expertise.

This paper has received reviews from reviewers but some have not provided a response to the author rebuttal. Given the limited time we have for author-reviewer discussions, we kindly ask you to share your post-rebuttal feedback to help clarify your perspective and aid the decision-making process.

Your input is invaluable in ensuring a fair and thorough review process.

Best,
AC

---

### Author Response · Authors · 2025-11-28
**Official Comment by Authors**

**Summary of Rebuttals and Gratitude**

We **sincerely thank the Area Chair and all Reviewers** for your time, constructive criticism, and patience throughout the rebuttal process. Your insightful review has significantly strengthened the quality, clarity, and scope of our work.

We would like to summarize the key improvements and clarifications made during this phase:

**1. Generalization to Vision-Language Tasks & nDCG Validation**

Addressing concerns about generalization and the definition of modality imbalance in retrieval (Reviewers SQuR, EPQt, L96f)

* We extended our evaluation to the Text-Vision Retrieval task (MSCOCO).
* We introduced nDCG@10, a ranking-aware metric, to quantitatively prove the existence of representation-level modality imbalance. The results show that CAMDrop consistently improves ranking quality, confirming that correcting encoder imbalance leads to better cross-modal alignment.

**2. Comparison with Recent SOTA Methods**

Responding to the request for more up-to-date comparisons (Reviewers sYcv, L96f)

* We added comparisons against recent state-of-the-art methods, including GMC (2022), DI-MML (2024), and LFM (2024). CAMDrop outperforms these methods on standard benchmarks while maintaining a simpler, plug-and-play design.

**3. Comprehensive Ablation Studies**

To validate our design choices (Reviewer SQuR), We conducted a series of **new ablation experiments**, including:

* Masking single vs. both modalities;
* Computing GradCAM from unimodal vs. multimodal logits;
* Per-instance vs. batch-averaged dominance ratio computation. These results confirm the optimality of our current configuration.

**4. Quantitative Overhead Analysis**

Addressing concerns regarding the "lightweight" claim (Reviewers sYcv, SQuR)

* We provided a detailed analysis of training overhead. The results show that CAMDrop introduces **negligible cost (~0.25% increase in wall-clock time per epoch)** and no increase in theoretical FLOPs, reaffirming its efficiency for real-world training.

**5. Investigation of Randomness vs. Semantic Guidance**

To clarify the necessity of GradCAM (Reviewer L96f, SQuR)

* We compared CAMDrop against four distinct random strategies: (1) Random Spatial Masking, (2) Random Channel Masking, (3) Hybrid Masking, and (4) Deterministic CAMDrop.
* The results empirically prove that semantic, class-relevant masking is essential for effective modality balancing, significantly outperforming random approaches.

**6. Clarification on Theoretical Concepts**

We provided detailed explanations for theoretical question:

* Theoretical Mechanism (Reviewer sYcv): We provided a formal analysis of how GradCAM-based masking redistributes gradients. By suppressing the "shortcut" regions of the dominant modality, CAMDrop effectively increases the relative learning signal for weaker modalities and enhances the mutual information between weak-modality features and labels
* Novelty & Interpretability (Reviewer L96f): We clarified the clear distinction between CAMDrop and prior methods (e.g., OPM, OGM-GE), emphasizing our shift from coarse, global dropping to fine-grained, class-specific spatial suppression. We also elaborated on how CAMDrop offers superior interpretability through semantic consistency and stability across instances.

**We remain fully engaged in the discussion and sincerely welcome any further feedback or questions from the reviewers during the remaining rebuttal period. We will continue to refine our manuscript based on your valuable inputs and ensure the final revised version is uploaded before the deadline. Thank you once again for your dedication and support!**

---

### Meta-Review · Area_Chair_uRQi · 2026-01-07

**Summary:**

reviewers gave scores of 2,4,4,4, highlighting concerns including:

The claim of CAMDrop being "lightweight" appears to be inaccurate and is a major weakness. The method requires generating a GradCAM heatmap for each modality on every training iteration and omits any analysis (e.g., training time, FLOPs) of this significant overhead.

Novelty beyond prior work in feature attribution is incremental.

Experimental concerns: certain hyperparameters are very sensitive (eg batchsize), reducing the "plug-and-play" nature of the method. Some baselines are outdated, needs comparisons on more datasets, and generalization to more modalities.

Absence of theoretical justification for why GradCAM-based masking should solve modality imbalance. The authors claim that "GradCAM identifies class-relevant spatial features and adaptively masks them based on instance-level importance," but provide no analysis showing how this directly addresses the core problem of modality imbalance.

Lack of qualitative analyses to demonstrate importance of interpretability.

Lack of ablation studies to analyze various design choices.

**Reviewer Concerns:**

The claim of CAMDrop being "lightweight" appears to be inaccurate and is a major weakness. The method requires generating a GradCAM heatmap for each modality on every training iteration and omits any analysis (e.g., training time, FLOPs) of this significant overhead.

--> authors added some experiments showing small computational overheads, but they still exist.

Novelty beyond prior work in feature attribution is incremental.

--> subjective concern, but seems like not addressed.

Experimental concerns: certain hyperparameters are very sensitive (eg batchsize), reducing the "plug-and-play" nature of the method. Some baselines are outdated, needs comparisons on more datasets, and generalization to more modalities.

--> authors added some baselines GMC (2022), DI-MML (2024), and LFM (2024), but these still seem quite outdated. also added a text-vision retrieval task. no discussion of hyperparameter sensitivity. partially addressed.

Absence of theoretical justification for why GradCAM-based masking should solve modality imbalance. The authors claim that "GradCAM identifies class-relevant spatial features and adaptively masks them based on instance-level importance," but provide no analysis showing how this directly addresses the core problem of modality imbalance.

--> authors provided some theoretical justification but it does not seem particularly deep or thorough. since this is a main concern it should have been addressed in the original version of the paper.

Lack of qualitative analyses to demonstrate importance of interpretability.

--> not addressed.

Lack of ablation studies to analyze various design choices.

--> authors added masking single vs. both modalities; computing GradCAM from unimodal vs. multimodal logits; per-instance vs. batch-averaged dominance ratio computation... seems addressed.

**Reviewer Scores:**

given the common concerns highlighted by all reviewers i think they are unlikely to significantly change their scores.

---

### Decision · Program_Chairs · 2026-01-26

Reject